# Categorical Distributions are Effective Neural Network Outputs for Event Prediction

## Abstract

We demonstrate the effectiveness of using a categorical distribution as a neural network output for the task of next event prediction. We find that training set sizes help explain performance differences between models: when training sets are increased, performance differences largely disappear. We introduce 3 new datasets which provide informative ways to explore model performance; they demonstrate cases where larger models and the use of the categorical output are effective.

## 1 Introduction

Probabilistic modeling of event data with neural networks is the concern of the field of neural network based temporal point processes (TPPs), reviewed by Shchur et al. (2021), Bosser & Taieb (2023) and Lin et al. (2025).

A variety of continuous distributions have been used as neural network outputs for predicting future event times (Bosser & Taieb, 2023). In this work, we propose to discretize the output domain, roughly into equal-quantile intervals, and to use the output of a neural network to represent a categorical distribution over these intervals. This approach is motivated by the observation that many datasets collected by measuring real-world processes exhibit a mixed distribution containing continuous and discrete parts. For example, the times between taxi pickups in New York City obtained by Whong (2014) have a distribution (see Figure 1) that is relatively smooth but punctuated by regular peaks. This type of distribution is not uncommon, and it will be shown that a categorical output interpreted as a piecewise constant distribution is effective at modeling such data.

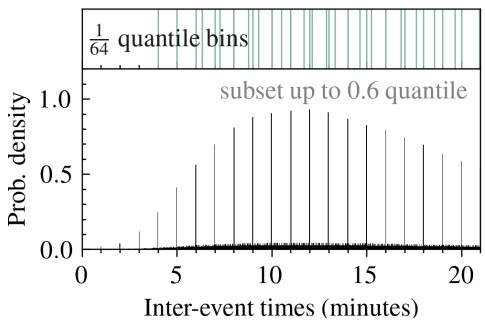

Figure 1: **Bottom**: Histogram of inter-event times for the NYC taxi dataset (Whong, 2014). **Top**: Intervals containing $\frac{1}{64}$ of events—these intervals are mappable to a categorical distribution with 64 outcomes.

The categorical output is not universally effective (nor is any output tested). Bosser & Taieb (2023) cataloged the performance differences of many neural TPP models, but did not find an explanation for these differences. In Section 4, we show that for many models and datasets, performance differences diminish when training set sizes are increased, suggesting that factors such as regularization contribute to performance differences at commonly used (and relatively small) training set sizes. To facilitate this investigation, we extend several existing datasets to larger sizes (see Figure 2).

We then introduce 3 new datasets which fill gaps in the existing dataset landscape. The synthetic datasets tested in Section 4 are shown to quickly plateau in terms of model performance gains they yield as training data is increased. In Section 5, we introduce a synthetic dataset using a modified Metropolis-Hastings algorithm that continues to yield performance gains for orders of magnitude larger training set sizes. This is useful for representing processes whose underlying dynamics are not quickly uncovered. In Section 6, we introduce a neuronal spike prediction task motivated by retinal prosthetics. For this dataset, not only are events from a real-world process, the task itself replicates the requirements of a real-world task. This is a feature lacking from the real-world datasets studied in Section 3 and Section 4. Finally, motivated by the structure of the spike prediction task, we introduce a family of synthetic datasets with discrete event times in Section 7. These datasets

Table 1: Two models evaluated in terms of test set NLL (lower is better) on existing real-world datasets. The `rnn-logmix` model from Shchur et al. (2020) acts as a baseline. The `rnn-cat` model uses the same architecture but with a categorical output structure. Values are means over 10 trials. Results for `rnn-cat` are reported by their difference to the results for `rnn-logmix`.

| Dataset | rnn-logmix | rnn-cat |
|---------|-----------|---------|
| Yelp airport | 4.700 | +0.057 |
| Yelp Mississuaga | 3.890 | +0.034 |
| Twitter | 3.963 | +0.055 |
| Taobao | 2.409 | -0.553 |
| Wikipedia | 5.120 | +0.024 |
| Yelp Toronto | 4.884 | -0.014 |
| PUBG | 1.585 | -3.781 |
| MOOC | 1.862 | -2.326 |
| Reddit AskScience | 5.643 | -0.167 |
| Amazon | 5.579 | +0.029 |
| Reddit Politics | 4.628 | -0.721 |
| Last.fm | 5.069 | -0.014 |

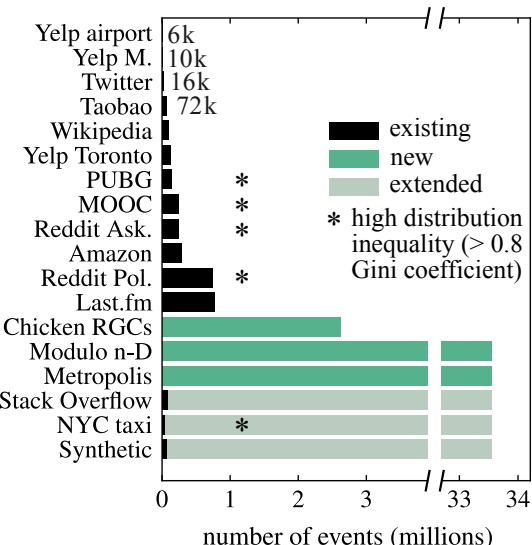

Figure 2: Dataset training set sizes. Datasets marked by ∗ have high Gini coefficient (> 0.8), which may indicate a discrete component in the distribution (see Appendix B). New and extended datasets are introduced in Sections 4 to 7.

reflect the discrete nature of many real-world datasets—discrete, for example, on account of the measurement process. For all of these new datasets, we observe the effectiveness of the categorical output. Compared to the existing datasets, the new datasets also demonstrate cases where larger models are effective.

The next section describes how a categorical distribution over intervals can be used for event prediction on the interval $(0, \infty)$.

## 2 CATEGORICAL DISTRIBUTION FOR INTER-EVENT TIMES ON $(0, \infty)$

For many event datasets, there is no maximum inter-event time, and models are expected to output distributions over the interval $(0, \infty)$. In this work, to form a continuous distribution on $(0, \infty)$, the categorical output is interpreted as assigning probability mass to intervals. The resulting distribution is a piecewise-continuous one, constructed as follows. We follow a similar approach to Kvamme & Borgan (2021) and choose intervals that equally divide the training set distribution: we fix $N$, the number of intervals, and choose interval lengths such that the first interval extends to contain the first $1/N$ of the inter-event times, the second until $2/N$ and so on until the last interval which extends to infinity. We let the probability *density* in each non-final interval be constant, determined by the interval width and the model's output. The final interval, being infinite in length, cannot have a constant non-zero density; instead we use an exponential decay weighted by the model's output. A subset of the intervals for a model with 64 outputs for the NYC taxi dataset is shown in Figure 1 (**top**). The intervals are fixed and do not change during training. Zero-width intervals are avoided by imposing a minimum interval length. See Appendix C for more details.

## 3 RESULTS ON EXISTING DATASETS

We compare the categorical output to the existing output heads listed in Table 2. Each output head is paired with the 3 model stems listed in Table 3. An additional model is also tested, the Transformer Hawkes Process, used in two sizes `thp-0` and `thp-1`, described by (Zuo et al., 2020). We evaluate the categorical output on the previously studied datasets listed in Figure 2 (excluding those that are expanded in the next section).

Table 2: Output heads used in experiments. See Appendix C for implementation details. To compare to the categorical head, focus is given to the `logmix` head throughout this work, as it is found to be the most effective among the existing heads tested.

| Label | Head description | Studied by |
|---|---|---|
| **cat** | categorical distribution | this work |
| const | exponential hazard | Huang et al. (2019), Li et al. (2018) |
| exp | constant hazard | Du et al. (2016), Upadhyay et al. (2018) |
| **logmix** | lognormal mixture (64 components) | Shchur et al. (2020) |
| nn | neural net parameterized hazard | Omi et al. (2019) |

Table 3: Model stems used in experiments. See Appendix C for implementation details. All existing heads from Table 2 were original studied with a recurrent neural network (RNN) stem.

| Label | Stem description | Input | Layers | Heads | Embed dim. | Parameters |
|---|---|---|---|---|---|---|
| rnn | gated recurrent unit | 32 | 1 | NA | 64 | 13k |
| gpt-a | GPT-2 transformer | 128 | 2 | 4 | 16 | 108k |
| gpt-b | GPT-2 transformer | 128 | 6 | 4 | 32 | 1.20M |

A subset of results is shown in Table 1, comparing `cat` and `logmix` heads using the `rnn` stem in terms of negative log-likelihood (NLL). Full results in terms of both NLL and mean absolute error (MAE) for all stem and head combinations are reported in Appendix A (Table 5 and Table 6).

The results in Table 1 show that the categorical output is competitive, although not universally more effective than the `logmix` output. The results, when viewed next to Figure 2, hint that dataset size or the presence of discrete components in the event distributions may be factors that help explain the results. The next section helps to understand the performance differences by evaluating over a range of training set sizes.

## 4 TRAINING SET SIZES DIFFERENTIATE TPP MODEL PERFORMANCE

From the existing real-world datasets listed in Figure 2, we identified the New York City taxi dataset (Whong, 2014; Du et al., 2016) and the Stack Overflow badge dataset (Du et al., 2016) as having data sources that allow the datasets to be recreated with much larger sizes. The 7 synthetic datasets from Omi et al. (2019) can also be recreated with larger sizes (arbitrarily large). In this section, the training sets of these 9 datasets are varied across 16 sizes from $2^{10}$ to $2^{25}$ inter-event times (while validation and test sets are fixed at $2^{17}$).

### 4.1 RESULTS AND DISCUSSION

All output heads from Table 2 are tested with the `rnn` and `gpt-a` stems from Table 3. Additionally, the `cat` and `logmix` heads are tested with the larger `gpt-b` stem. Figure 3 reports NLL for 6 of the 9 datasets. Appendix A contains the complete results (NLL and MAE for all datasets).

The results support three claims.

First, across all datasets, training set size is an important factor explaining model performance. For most datasets, the larger models take more samples to plateau but eventually catch up to or surpass their smaller counterparts. The sensitivity to training set size is also dependent on output structure. The exponential intensity head sees little difference across training set sizes, whereas the categorical output sees a large variation. This supports the claim that different output structures can be thought of as conferring different degrees of regularization, preventing performance degradation at smaller training set sizes. Across all datasets, in the presence of sufficient data, both the use of the categorical output and the use of larger models are competitive.

Second, the categorical output achieves strong performance on the NYC taxi dataset, irrespective of model and training set size. This is consistent with the hypothesis that the discrete nature of

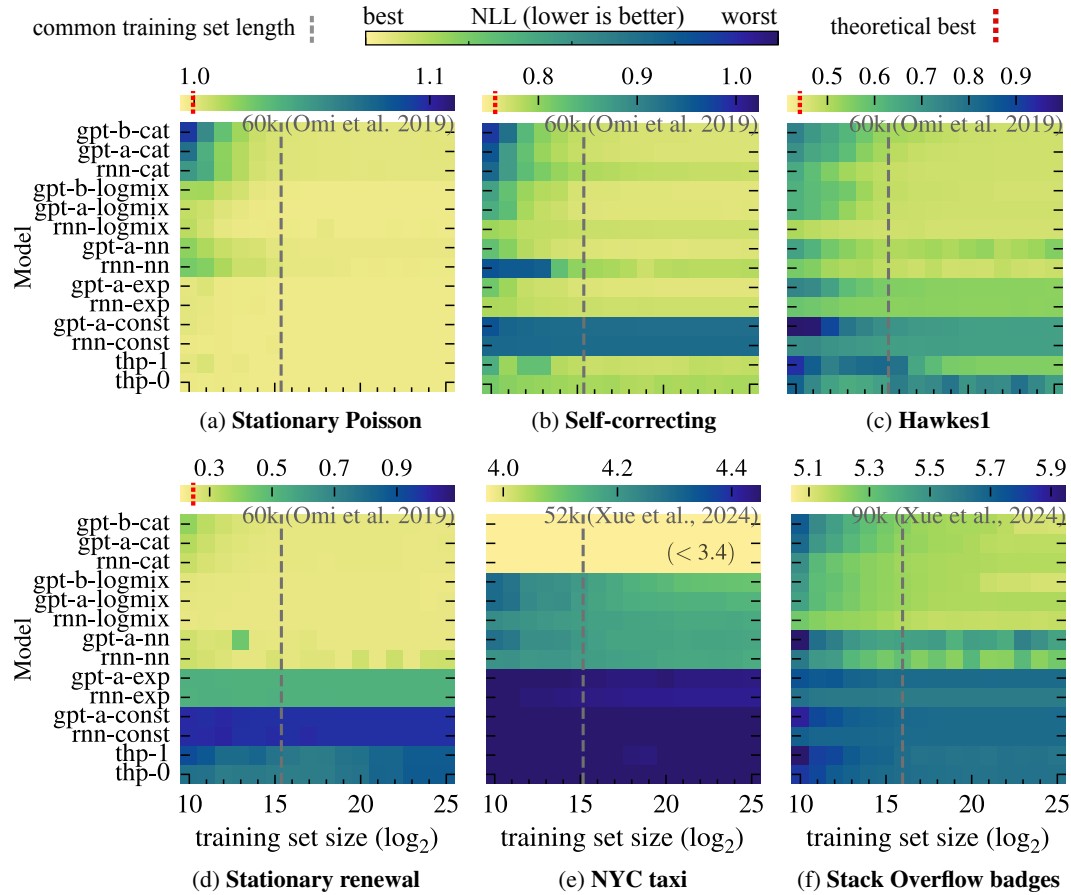

Figure 3: Performance comparison of 13 models in terms of test set NLL across 6 datasets and 16 training set lengths from $2^{10}$ to $2^{25}$. The colormap ranges span each sub-figure's full range of values, except for the NYC taxi dataset, where the categorical models' very low NLL scores (all $< 3.4$) are separated to preserve the colormap detail for other models. For synthetic datasets where a theoretical best score is known, it is marked with a red dashed line.

this dataset (see Figure 1 and Appendix B) explains the effectiveness of the categorical output. The `logmix` model can, in principle, represent distributions with sharp peaks; however, across all training set lengths, the `logmix` models are unable to match the performance of the categorical models. One hypothesis is that singularities hinder stable training of a lognormal mixture, similar to the situation encountered when fitting such mixtures with the expectation-maximization algorithm (Bishop, 2006); supporting evidence for this theory is presented in Appendix C.5.2.

Finally, a striking observation is that for all *synthetic* datasets there is a model that requires very little data to be a top-scoring model. We argue that this is evidence that the event sequences do not have complex dynamics needing significant training sets to learn. Take for example the stationary Poisson process where, after a few thousand samples, most of the models reach close to the theoretical best NLL score, and further improvement would effectively require inferring the state of the underlying pseudo-random number generator. In the next section, we introduce a new synthetic dataset that continues to yield performance gains over longer training lengths.

## 5 METROPOLIS LOGNORMAL EVENT DATASET

The previous section showed how the synthetic datasets from Omi et al. (2019) quickly plateau in terms of the performance gains they yield, explaining how small models can reach competitive

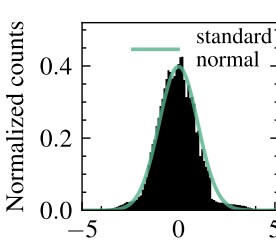

Figure 4: Histogram of 200k samples from the modified Metropolis-Hastings algorithm.

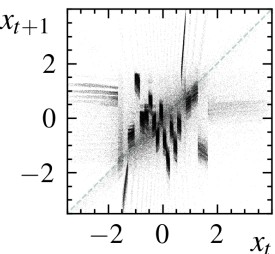

Figure 5: 200k pairs of samples rasterized: $x_{t+1}$ and previous sample $x_t$.

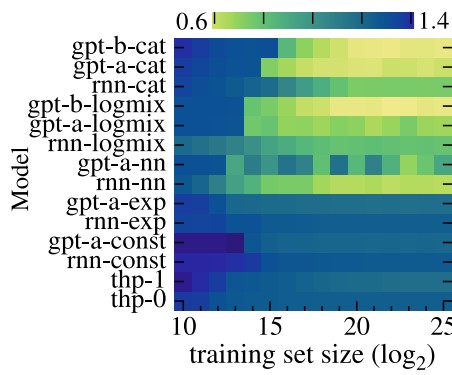

Figure 6: Test set NLL scores on the Metropolis lognormal dataset.

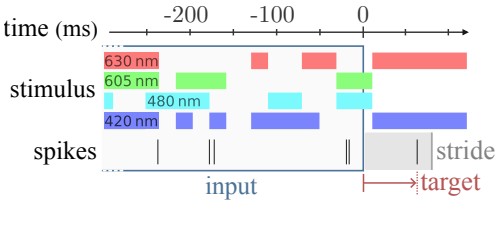
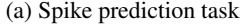

(a) Spike prediction task

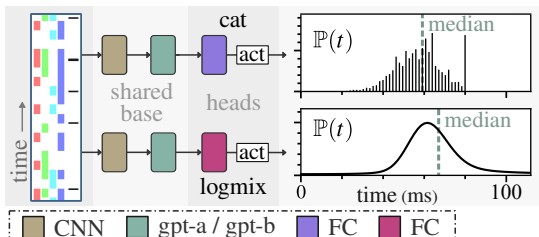

(b) Categorical and logmix heads

Figure 7: **(a)** Spike prediction task: given a 1-second snippet of stimulus and spike history, predict the time of the next spike. Predictions beyond a stride of $80\,\mathrm{ms}$ are not used. Information after $t = 0$ is not available to a model, matching the task faced by retinal prosthetic devices. **(b)** The inputs, architecture and outputs of the models tested. Figure adapted from Doran et al. (2024).

performance on small training sets. Here, we use a Markov process to create a synthetic dataset that continues to yield performance gains for orders of magnitude larger training set sizes.

We generate an event sequence from a nested set of state transition matrices. The matrices parameterize a modified Metropolis-Hastings sampling algorithm. This process exploits what is typically considered a weakness of Metropolis-Hastings sampling—that samples are dependent—in order to gradually leak information about the state transition matrices. The generation procedure is described in Appendix B.5. Figure 4 plots samples from the sequence before applying the logarithm, revealing the roughly Gaussian shape of the distribution. For the same samples, Figure 5 plots next vs. previous samples, highlighting that with enough samples, a highly structured distribution emerges.

Figure 6 shows how models perform across a range of training set lengths. Here, all heads (except `nn`) benefit from having a larger base model, and smaller models do not quickly approach a competitive score. This dataset is important for being a *synthetic* event dataset where there is a clear benefit of using larger models, and those models gradually improve with more data. Regarding `gpt-a-nn`: we observed this model to be more prone to training instability compared to `rnn-nn`, a possible explanation for its poorer performance.

## 6 A CATEGORICAL OUTPUT IS EFFECTIVE FOR SPIKE PREDICTION

The real-world datasets studied in previous sections are event sequences recorded from real processes; however, predicting the next event in these sequences is not grounded in a realistic task. This section introduces a task motivated by retinal prosthetic devices. We demonstrate how the demands of a real-world task can lead to a discrete output structure being suitable for predicting events of an otherwise continuous process. In this setting, we again see the usefulness of the categorical distribution.

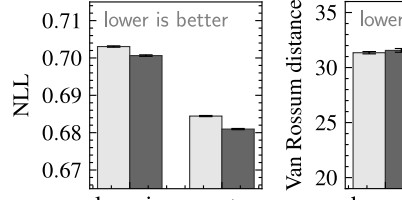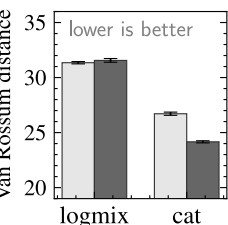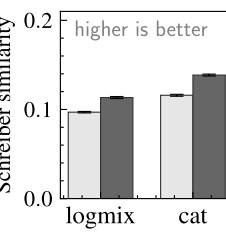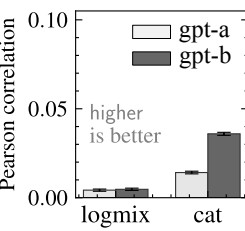

Figure 8: Performance comparison between *logmix* (a 64-component lognormal mixture) and *cat* (an 81-bin categorical distribution) for the task of spike prediction. NLL is reported on the left. The next 3 metrics are spike train similarity/distance metrics, evaluated autoregressively; they share a smoothing parameter of $60\,\text{ms}$. The model stem is tested in two sizes: `gpt-a` and the larger `gpt-b`. We follow Agarwal et al. (2021) and report metrics as interquartile means over the 1611 chicken RGCs over 10 training repeats, with 95% bootstrap confidence intervals as error bars.

Retinal prosthetic devices must mimic the activity patterns of cells by considering impinging light and previous spike activity. Gogliettino et al. (2023) describes such a device for electrically stimulating the primate retina. We design a task that captures some of the requirements of such a device.

We create a dataset of snippets from multi-electrode array recordings of chicken retinal ganglion cells (RGCs) responding to visual stimuli (Seifert et al., 2023). An input is a 1024-length snippet of a recording ( ~1 second), containing the stimulus and spike history of a single cell. The target is the time until the cell's next spike (see Figure 7a). There are 1611 cells across 16 recordings. Further details on the dataset are contained in Appendix D. To gauge how well models might operate in a retinal prosthetic device, we roll out the models autoregressively and compare the generated spike trains to the ground truth spike trains using common spike train similarity metrics.

## 6.1 SAMPLING AND PREDICTION RATES MOTIVATE A CATEGORICAL OUTPUT

Two temporal scales are associated with the task: the sampling period and the prediction period. The *sampling period* of the spikes and stimulus is ~1 ms ($1.008\,\text{ms}$). The autoregressive regime introduces the *prediction period*—the maximum duration to wait before making a new prediction. A shorter prediction period allows for faster integration of new stimulus information, but carries a computational cost. We fix 80 samples (~$80\,\text{ms}$) as the prediction period. There are diminishing returns to reducing this period, as $80\,\text{ms}$ is close to the mean response delay of the RGCs recorded by Seifert et al. (2023).

The two temporal scales make the categorical distribution a natural choice for a model output. With $80\,\text{ms}$ prediction period and $1\,\text{ms}$ sampling period, a categorical distribution with 81 outcomes is the minimal neural network output that remains fully descriptive: 80 classes for the 80 intervals in the prediction period and one class for the interval $[80\,\text{ms}, \infty)$. The sampling frequency implies that no finer resolution than $1\,\text{ms}$ is needed, and the prediction frequency ensures that no granularity at all is needed beyond $80\,\text{ms}$. Figure 7b shows the categorical output being used for the spike prediction task above the output head it is compared against, the mixture of lognormals.

## 6.2 MODELS: HEAD(CNN + TRANSFORMER)

We train 4 models. The `logmix` and `cat` heads are attached to the same architecture: a convolutional neural network (CNN) followed by `gpt-a` or `gpt-b`. A straightforward ResNet (He et al., 2016) based CNN stem encodes a 1024-length frame of stimulus and spike history into a 64-length sequence of vectors of length 64. This sequence is passed to `gpt-a` (or `gpt-b`) from Table 3. An embedding of the cell number is added to the input sequence. The output of the transformer is either an 81 length vector in the case of the categorical output, or a $64 \times 3 = 192$ length vector in the case of the mixture of 64 lognormals. Appendix C describes the models in more detail.

### 6.3 EXPERIMENT AND RESULTS

The models are trained at next spike prediction with a NLL loss (see Appendix D for training details). Point estimates are made using the median of the output distribution. We evaluate model performance with 4 metrics: NLL, Schreiber Similarity, Van Rossum distance and smoothed Pearson correlation. The latter 3 metrics are calculated using the spike trains generated by rolling out the models autoregressively; the metrics share a smoothing parameter set to $60\,\mathrm{ms}$, a duration shown to be appropriate by Doran et al. (2024). Schreiber similarity introduced by Schreiber et al. (2003) and Van Rossum distance introduced by Van Rossum (2001) are common metrics used to compare spike trains. See (Paiva et al., 2010) for a review of spike train comparison metrics. The NLL calculation uses probability mass, and for the logmix model, this involves integrating the continuous distribution over a $1\,\mathrm{ms}$ interval.

All 4 metrics are reported in Figure 8. The categorical output consistently outperforms the mixture of lognormals, highlighting the effectiveness of the simple output structure. The benefit of using the larger model stem (`gpt-b`) is also clear.

## 7 EVENT SEQUENCES FROM MODULO ADDITION

The spike prediction task justified the use of a discrete representation for events of a continuous process. This section introduces a family of synthetic processes that generate events in discrete intervals. The aim is for these synthetic processes to be a useful tool to investigate model performance in the same way that synthetic processes such as Hawkes processes are useful in the continuous setting. Modulo addition will be used to create the event sequences. The sequences can be related to real-world processes that involve wrap-around events, such as timer or frame counter overflows.

Using modulo addition, an increasing sequence defined over a grid can be used to generate event sequences with varying degrees of complexity. Consider tracking a particle along a 1D track and recording when the particle passes 10 meters, 20 meters, 30 meters and so on. Over a 2D plane, we could record the times when a particle moves from one quadrant to another. As the number of dimensions increases, the number of faces a particle can exit through to enter another division increases. Figure 10 shows a particle moving in 1D, 2D and 3D space and generating events with respect to a grid. To predict when a particle will move to the next quadrant, the previous events can be observed to narrow down the particle's position and velocity. This becomes more challenging and requires more data points as the number of dimensions increases. By varying the number of dimensions, a family of event sequences can be created with various decoding difficulties.

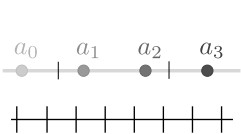 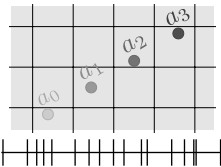 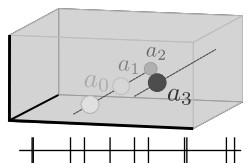

Figure 9: Four snapshots of a particle moving in 1, 2 and 3-dimensional space. When the particle passes a boundary, an event is generated. An example event sequence is shown. The 3-dimensional case is visualized by the particle wrapping around in a single voxel.

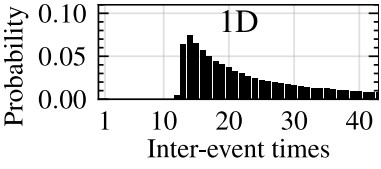 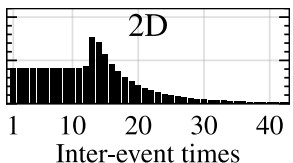 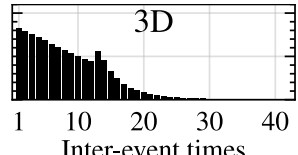

Figure 10: Inter-event time distributions for 1, 2 and 3-d modulo sequences where the starting position, $a_0$, is sampled uniformly from $[0, 1021)^d$ and the velocity, $v$, uniformly from $[10, 80]^d$.

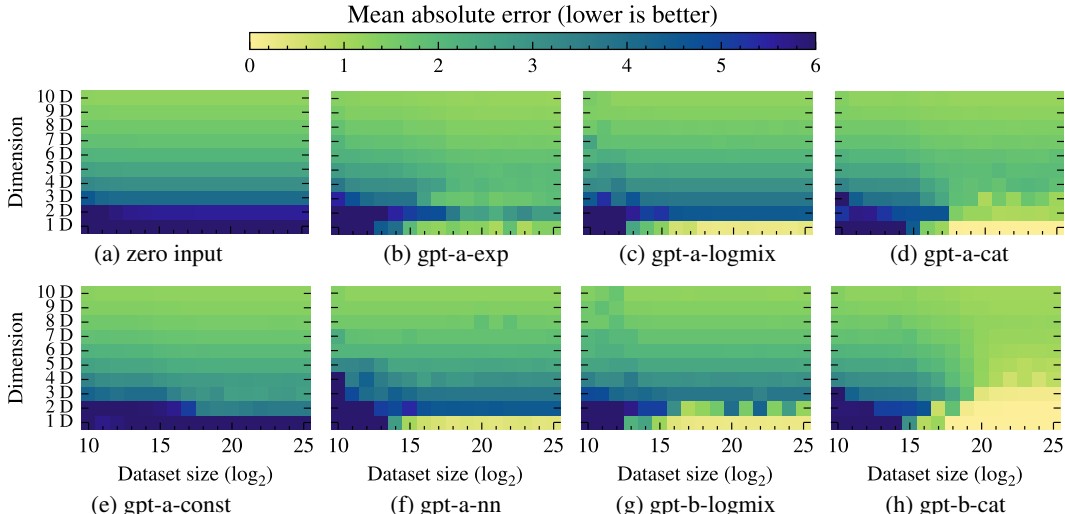

Figure 11: Performance comparison of 8 models in terms of test set mean absolute error (MAE). Each model is trained and evaluated over a landscape of 16 training set sizes for 10 modulo datasets. Model (a) takes no input and just outputs the empirical training set distribution; it acts as a baseline.

## 7.1 10 DATASETS, FROM 1D TO 10D

We create 10 datasets of *discrete time* event sequences, one for each dimension from 1 to 10.

**A 1-dimensional sequence** is generated as follows. A grid size $n$ is fixed. In this work, we use $n = 1021$. A velocity $v$ and starting position $a_0$ are chosen. A sequence of positions $a_0, a_1, a_2, \ldots$ is generated by $a_{i+1} = a_i + v$. Whenever $a_{i+1} < a_i \bmod n$, an event is generated at time $t = i+1$.

**A d-dimensional sequence** is generated from *vector* addition in the same way. A vector $\mathbf{n}$ of length $d$ is fixed. We fix $\mathbf{n}$ to be $\mathbf{n} = (1021, 1021, \ldots, 1021)$. A velocity $\mathbf{v}$ and starting position $\mathbf{a_0}$ are chosen. A sequence of positions $\mathbf{a_0}, \mathbf{a_1}, \mathbf{a_2}, \ldots$ is generated by $\mathbf{a_{i+1}} = \mathbf{a_i} + \mathbf{v}$. Whenever any of the components of $\mathbf{a_i}$ overflow with respect to $\mathbf{n}$ when transitioning to $\mathbf{a_{i+1}}$, an event is generated at time $t = i+1$.

The 1D dataset is formed of many separate event sequences of length 1024 generated for different $a_0$ and $v$. To create a sequence, we sample $a_0$ uniformly from $[0, 1021)$ and $v$ uniformly from $[10, 80]$ and generate 1024 events per sequence. The other 9 datasets are generated in the same way. Having many short sequences rather than a single long sequence makes the dataset more representative of datasets like the NYC taxi dataset, the Stack Overflow dataset and the chicken RGC dataset, all of which contain numerous separate sequences. More details on the generation process, including motivations for the bounds of $v$ and an analogous continuous process, are described in Appendix B.7. For each dataset, we generate $2^{17}$ sequences, for a total of $2^{27}$ events per dataset. This is then split into training, validation and test sets in an 8:1:1 ratio. As before, we consider 16 subsets of the training set, containing $2^0, 2^1, ..., 2^{15}$ sequences ($2^{10}, 2^{11}, ..., 2^{25}$ events).

## 7.2 RESULTS AND DISCUSSION

Seven `gpt-a` and `gpt-b` based models are trained on the modulo datasets. Figure 11 shows MAE for the models across the landscape of modulo datasets. NLL scores appear in Appendix A. The first figure, Figure 11a, is the result of predicting using only the training set's empirical distribution (no event history input is used) and acts as a baseline for comparison.

The first observation is that for all dimensions, there is a training set size above which `gpt-a-cat` is equal to or better than the other `gpt-a` models. This result is another example demonstrating the competitiveness of the categorical head when there is sufficient data. We also see the benefit of using larger models when training set sizes are large, demonstrated by a reduction in MAE for logmix and categorical heads when switching from `gpt-a` to `gpt-b`. There are also differences

between models when using small training sets; for example, the logmix head is more effective than the categorical head. This effect is relatively insensitive to model size, which is evidence that the output structures themselves affect generalization when data is limited.

By comparing to the zero input baseline, we see that the usefulness of the contextual information decreases with increasing dimension—by 10D, even with $2^{25}$ samples, no model markedly outperforms the zero input baseline. From this perspective, the modulo datasets form a spectrum between easy (1D) and difficult (10D) in terms of inferring the next event from event histories. In Appendix B.7, the concept of dataset difficult is analysed in terms of $\mathcal{V}$-information introduced by Ethayarajh et al. (2022).

## 8 DISCUSSION

An argument of this work is that the task and dataset should drive model choice. In the spike prediction task, the task itself characterized a categorical output structure. The NYC taxi dataset, whether due to the discrete nature of the recording process or the presence of periodicity in the underlying system, has a distribution of inter-event times with many narrow peaks and is conducive to being represented with a categorical distribution. This situation is not uncommon—the histograms of all real-world datasets used in this work are listed in Appendix B, and many have shapes with narrow peaks capturing the majority of events. In addition to the inescapable discrete nature of many event sequences recorded from real-world processes, realistic tasks associated with these sequences may also define a resolution of interest. The spike prediction task is one such example. It is reasonable to expect that use cases for predicting the next taxi request or the next visit to a location would also describe a resolution at which predictions are relevant, such as seconds or minutes. Evaluating probability mass within intervals would call into question the density-based loss signals typically used for training TPP models. It would be interesting for future works to consider what other output representations may be effective if mass rather than density based evaluations are used.

A second theme of this work is explaining performance differences between models. Section 4 showed how the training set sizes used in existing benchmarks snapshot the performance at a point where reducing model size can be beneficial, presumably on account of the regularization effect of model capacity. It was also shown that the synthetic datasets from Omi et al. (2019) are such that small models achieve competitive performance at short training set sizes. This was argued to be on account of how quickly these datasets plateau in terms of the performance gains they afford as training data is increased. Both of these points suggest that existing benchmarks may be of limited use to practitioners interested in using large models to exploit large amounts of data.

## 9 LIMITATIONS

Several limitations reduce the strength of the conclusions. The work does not consider the prediction of additional spatial or categorical information. The exposition of both discrete and continuous settings makes the treatment more comprehensive, but makes it more difficult to compare results across sections. This work broadly ascribes model capacity and output structure as being important for explaining performance across datasets; however, specific mechanisms are not isolated, such as data memorization or the alignment of the output distribution family to a dataset's distribution.

## 10 CONCLUSION

This work demonstrates that the categorical distribution is an effective representation for spike prediction across a range of datasets. On existing TPP datasets, we show that many performance differences between models reduce as training set sizes are increased. We show that for existing synthetic datasets, model size has very little effect on performance. We introduce a synthetic dataset where larger models can make use of larger training set sizes. We describe a case study (neuronal spike prediction) where the categorical distribution is a natural fit given the structure of the task. This task is additionally valuable as it is accompanied by application-relevant performance metrics. Inspired by the spike prediction task, we introduce a family of synthetic datasets where events are recorded in finite intervals.

REPRODUCIBILITY

Supplementary code (including datasets) permits all experiments to be reproduced. The Stack Overflow dataset is not included, as the usage agreement prevents distribution; for this case, instructions for obtaining the data are provided in Appendix B.3. To accompany the code, the appendix also includes descriptions of models (Appendix C), datasets (Appendix B) and training procedures (Appendix E).

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

## A  SUPPLEMENTARY RESULTS

This appendix presents results supplementary to Sections 3 to 7.

Table 1 from Section 3 (*Results on existing datasets*) showed results for `rnn-logmix` and `rnn-cat` evaluated on 12 real-world datasets. These results are extended by Table 4 to include information on the variability over the 10 runs. Many of these datasets have fixed training, validation and test sets, so training runs do not differ in terms of data used, only in terms of initialization and training stochasticity.

In Table 5, the same 12 datasets are used to test all model combinations of `rnn`, `gpt-a` and `gpt-b` stems with heads `const`, `exp`, `nn`, `logmix` and `cat`, along with the Transformer Hawkes models, `thp-0` and `thp-1`. Table 6 shows the corresponding MAE scores.

In Section 4 (*Training set sizes differentiate TPP model performance*), Figure 3 showed NLL scores on a subset of the synthetic datasets used in this work. Figure 12 and Figure 13 below show the complete set of NLL and MAE scores for the synthetic datasets, and Figure 14 and Figure 15 show the NLL and MAE scores for the real-world datasets.

In Section 7 (*Event sequences from modulo addition*), results were reported in terms of MAE. The corresponding NLL scores are shown in Figure 16.

Table 4: Extension of Table 1 from Section 3 (*Results on existing datasets*). Two models (`rnn-logmix` and `rnn-cat`) evaluated in terms of test set NLL (lower is better) on existing real-world datasets. Point estimates are the mean over 10 trials. Variability is expressed as 95% confidence intervals calculated assuming a normal distribution of the mean: $\pm 1.96 \frac{s}{\sqrt{10}}$, where $s$ is the sample standard deviation.

| Dataset | rnn-logmix | rnn-cat |
|---|---|---|
| Yelp airport | 4.700 ± 5.43e-03 | 4.756 ± 2.63e-03 |
| Yelp Mississuaga | 3.890 ± 1.41e-03 | 3.925 ± 1.68e-03 |
| Twitter | 3.963 ± 1.60e-03 | 4.017 ± 1.42e-03 |
| Taobao | 2.409 ± 8.53e-04 | 1.856 ± 3.12e-04 |
| Wikipedia | 5.120 ± 1.07e-03 | 5.144 ± 7.58e-04 |
| Yelp Toronto | 4.884 ± 1.05e-02 | 4.870 ± 3.81e-04 |
| PUBG | 1.585 ± 7.33e-02 | -2.195 ± 5.43e-04 |
| MOOC | 1.862 ± 9.01e-02 | -0.464 ± 1.19e-03 |
| Reddit AskScience | 5.643 ± 1.49e-02 | 5.476 ± 3.54e-04 |
| Amazon | 5.579 ± 9.28e-03 | 5.608 ± 2.48e-04 |
| Reddit Politics | 4.628 ± 4.01e-02 | 3.906 ± 4.84e-04 |
| Last.fm | 5.069 ± 1.56e-02 | 5.055 ± 4.94e-03 |

### A.1  DISCUSSION

The supplementary results motivate the discussion of two points: the contrast between MAE and NLL metrics, and the presence of infinite NLL scores.

The MAE scores on the synthetic and real-world datasets (Figure 13 and Figure 15) show a similar pattern to the NLL scores (Figure 12 and Figure 14). Quick plateauing of performance on the synthetic datasets from Omi et al. (2019) is even more apparent in terms of MAE. The ability to concentrate probability mass allows the models with a categorical output to achieve a low NLL score on the NYC taxi dataset does not correspond to a similarly low MAE score. This is further evidence to support the claim that NLL should be evaluated in terms of probability mass by specifying an interval of interest, such as hours, rather than evaluating in terms of probability density. Indeed, a theoretically best scoring model in terms of NLL simply outputs arbitrarily large probability densities at the countable rationals, yet such a model would not score well in terms of other metrics such as MAE.

Table 5: Extension of Table 1 from Section 3 (*Results on existing datasets*). Test set NLL scores (lower is better) for various models on real-world datasets. 5 output heads (`logmix`, `cat`, `nn`, `const` and `exp`) are paired with 3 model stems (`rnn`, `gpt-a` and `gpt-b`). Additionally, `thp-0` and `thp-1` from Zuo et al. (2020) are included. Columns are ordered in terms of training set length, from largest (left) to smallest (right). The `rnn-logmix` is taken as the baseline (first column), and all other results are reported as differences from these results. For the `logmix` and `cat` output heads, results are the mean over 10 runs, while all other results are from a single run.

| Model | Last.fm | Reddit Pol. | Amazon | Reddit Ask. | MOOC | PUBG | Yelp Tor. | Wikipedia | Taobao | Twitter | Yelp M. | Yelp airport |
|---|---|---|---|---|---|---|---|---|---|---|---|---|
| rnn-logmix | 5.069 | 4.628 | 5.579 | 5.643 | 1.862 | 1.585 | 4.884 | 5.120 | 2.409 | 3.963 | 3.890 | 4.700 |
| gpt-b-cat | -0.045 | -0.744 | +0.045 | -0.161 | -2.267 | -3.797 | -0.263 | +0.104 | -0.528 | +0.141 | +0.057 | +0.081 |
| gpt-a-cat | -0.101 | -0.746 | +0.038 | -0.166 | -2.291 | -3.801 | -0.004 | +0.082 | -0.536 | +0.122 | +0.035 | +0.061 |
| rnn-cat | -0.013 | -0.721 | +0.029 | -0.167 | -2.326 | -3.780 | -0.014 | +0.024 | -0.553 | +0.054 | +0.034 | +0.057 |
| gpt-b-logmix | -0.068 | -0.829 | +0.125 | -0.705 | -0.122 | +0.344 | -0.264 | +0.079 | +0.019 | +0.076 | +0.001 | +0.005 |
| gpt-a-logmix | -0.115 | -0.959 | +0.078 | -0.941 | -0.608 | +0.459 | -0.210 | +0.065 | +0.012 | +0.075 | -0.003 | +0.001 |
| gpt-b-nn | +0.028 | +0.468 | +0.079 | +0.184 | +1.768 | +2.323 | +0.092 | +0.163 | +0.026 | +0.147 | +0.078 | +0.068 |
| gpt-a-nn | +0.035 | +0.466 | +0.048 | +0.167 | +1.647 | +1.807 | +0.163 | +0.088 | +0.017 | +0.084 | +0.017 | +0.028 |
| rnn-nn | +0.227 | +0.479 | +0.097 | +0.163 | +1.632 | +2.225 | +0.105 | +0.055 | +0.002 | +0.030 | +0.048 | +0.022 |
| gpt-b-exp | +1.109 | +0.482 | +2.704 | +0.173 | +4.158 | +2.400 | +0.340 | +1.008 | +0.301 | +0.374 | +0.045 | +0.025 |
| gpt-a-exp | +1.173 | +0.492 | +2.713 | +0.173 | +4.165 | +2.398 | +0.340 | +0.973 | +0.299 | +0.376 | +0.039 | +0.016 |
| rnn-exp | +1.299 | +0.488 | +2.702 | +0.166 | +4.159 | +2.402 | +0.336 | +0.919 | +0.290 | +0.325 | +0.046 | +0.008 |
| gpt-b-const | +1.517 | +0.483 | +2.935 | +0.189 | +5.601 | +2.410 | +0.356 | +1.605 | +0.663 | +0.612 | +0.063 | +0.045 |
| gpt-a-const | +1.505 | +0.483 | +2.935 | +0.186 | +5.600 | +2.410 | +0.355 | +1.600 | +0.671 | +0.606 | +0.062 | +0.039 |
| rnn-const | +1.541 | +0.486 | +2.933 | +0.180 | +5.679 | +2.413 | +0.351 | +1.496 | +0.648 | +0.574 | +0.069 | +0.026 |
| thp-1 | +1.338 | +0.484 | +2.717 | +0.201 | +5.120 | +2.401 | +0.364 | +1.461 | +0.317 | +0.480 | +0.046 | +0.019 |
| thp-0 | +1.456 | +0.491 | +2.713 | +0.192 | +4.224 | +2.401 | +0.370 | +1.094 | +0.309 | +0.377 | +0.039 | +0.018 |

Table 6: MAE version of Table 1 from Section 3 (*Results on existing datasets*). Test set MAE scores (lower is better) for various models on real-world datasets. 5 output heads (logmix, cat, nn, const and exp) are paired with 3 model stems (rnn, gpt-a and gpt-b). Additionally, thp-0 and thp-1 from Zuo et al. (2020) are included. Columns are ordered in terms of training set length, from largest (left) to smallest (right). The rnn-logmix is taken as the baseline (first column), and all other results are reported as differences from these results. For the logmix and cat output heads, results are the mean over 10 runs, while all other results are from a single run.

| Model | Last.fm | Reddit Pol. | Amazon | Reddit Ask. | MOOC | PUBG | Yelp Tor. | Wikipedia | Taobao | Twitter | Yelp M. | Yelp airport |
|---|---|---|---|---|---|---|---|---|---|---|---|---|
| rnn-logmix | 1325 | 51.79 | 861.9 | 194.5 | 731.6 | 18.33 | 64.97 | 412.2 | 7.502 | 36.57 | 20.17 | 34.46 |
| gpt-b-cat | +006 | -0.27 | -5.9 | +1.9 | +-0.0 | -0.97 | +0.06 | +4.6 | +0.051 | +1.06 | +0.46 | +1.16 |
| gpt-a-cat | +004 | -0.33 | -6.2 | +1.7 | -0.1 | -0.96 | +0.70 | +4.3 | +0.048 | +0.95 | +0.29 | +0.92 |
| rnn-cat | +002 | -0.23 | -11.0 | -0.6 | -0.4 | -0.91 | -0.04 | +0.7 | +0.012 | +0.33 | +0.38 | +0.35 |
| gpt-b-logmix | +006 | -0.20 | -6.1 | +0.6 | +-0.0 | -0.93 | +0.03 | +4.4 | +0.039 | +0.85 | +0.30 | +1.00 |
| gpt-a-logmix | +003 | +0.11 | -4.9 | +2.5 | +-0.0 | -0.93 | -0.28 | +4.1 | +0.029 | +0.89 | +0.20 | +0.89 |
| gpt-b-nn | +018 | -0.37 | +20.8 | +4.2 | +0.3 | -0.94 | +0.29 | +3.5 | +0.021 | +0.54 | +0.62 | +0.61 |
| gpt-a-nn | +024 | -0.23 | +20.7 | +3.8 | +0.3 | -0.89 | +0.93 | +4.5 | +0.024 | +0.50 | +0.43 | +0.81 |
| rnn-nn | +005 | -0.21 | +21.1 | -0.4 | +2.0 | -0.87 | +0.27 | +0.7 | -0.004 | +0.06 | +0.49 | +0.43 |
| gpt-b-exp | +065 | -0.35 | +185.8 | +1.3 | +27.6 | -0.96 | +0.30 | +24.4 | +0.185 | +1.34 | +0.42 | +0.62 |
| gpt-a-exp | +057 | +0.04 | +202.9 | +1.1 | +26.8 | -0.99 | +0.38 | +21.3 | +0.145 | +1.59 | +0.29 | +0.29 |
| rnn-exp | +078 | -0.22 | +190.4 | +-0.0 | +32.3 | -0.89 | +0.08 | +19.9 | +0.152 | +1.17 | +0.30 | +0.24 |
| gpt-b-const | +243 | -0.35 | +435.2 | +3.1 | +349.0 | -0.81 | +0.63 | +136.7 | +1.323 | +6.70 | +0.79 | +1.64 |
| gpt-a-const | +219 | -0.35 | +444.5 | +3.5 | +376.2 | -0.80 | +0.37 | +143.0 | +1.539 | +6.32 | +1.03 | +0.94 |
| rnn-const | +234 | -0.18 | +424.8 | +1.8 | +362.5 | -0.77 | +0.32 | +108.8 | +1.251 | +5.19 | +0.42 | +0.57 |
| thp-1 | +077 | +21.03 | +1002.7 | +57.5 | +0.2 | +4.52 | +18.11 | +10.5 | +0.431 | +3.74 | +5.22 | +10.17 |
| thp-0 | +077 | +20.98 | +1002.8 | +57.5 | +0.2 | +4.53 | +18.11 | +10.5 | +0.448 | +3.69 | +5.18 | +10.26 |

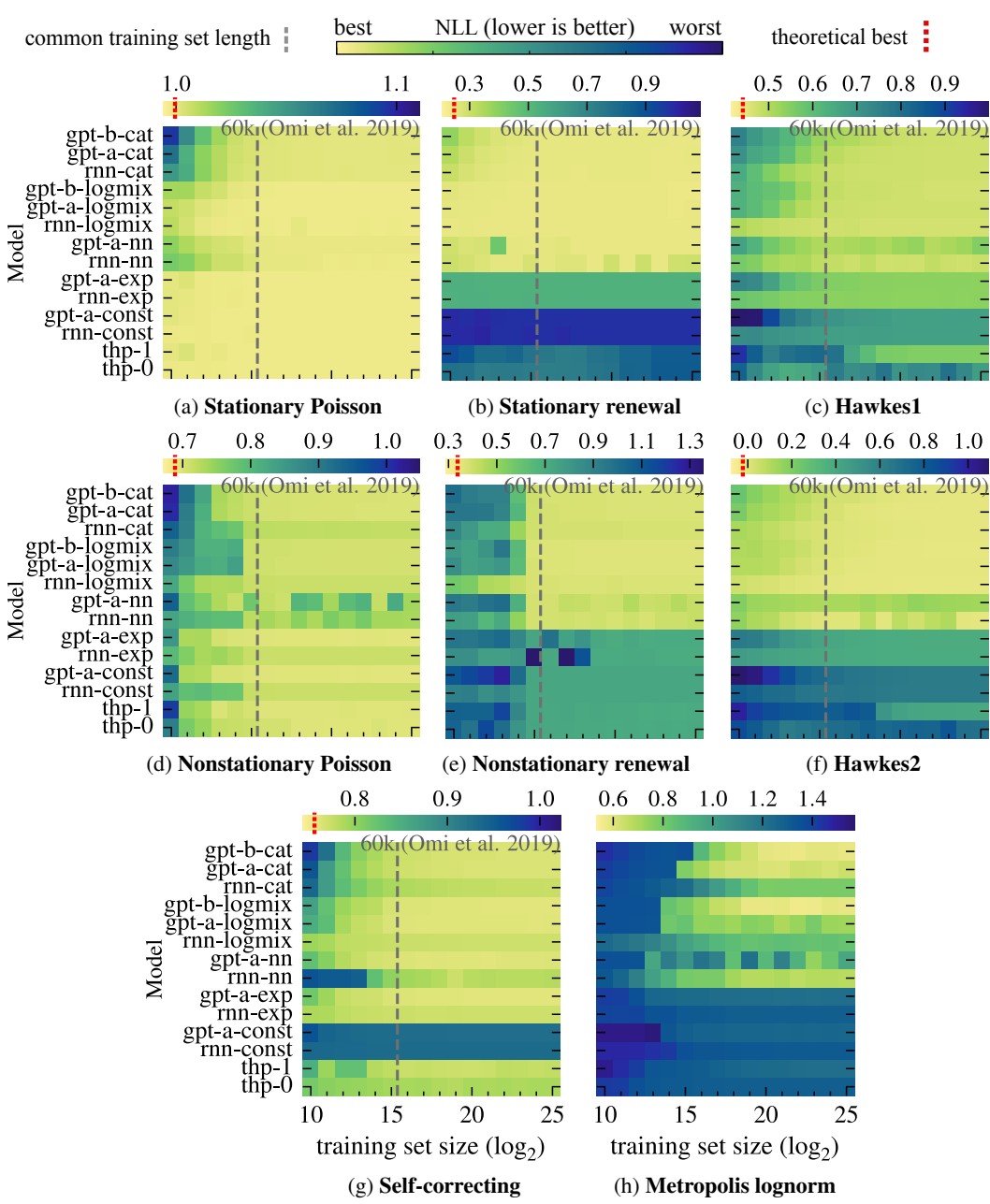

Figure 12: Performance comparison of 14 models in terms of test set NLL across 8 synthetic datasets. There are 16 training set lengths from $2^{10}$ to $2^{25}$. The colormap ranges span each sub-figure's full range of values. Where a theoretical best score is known, it is marked with a red dashed line.

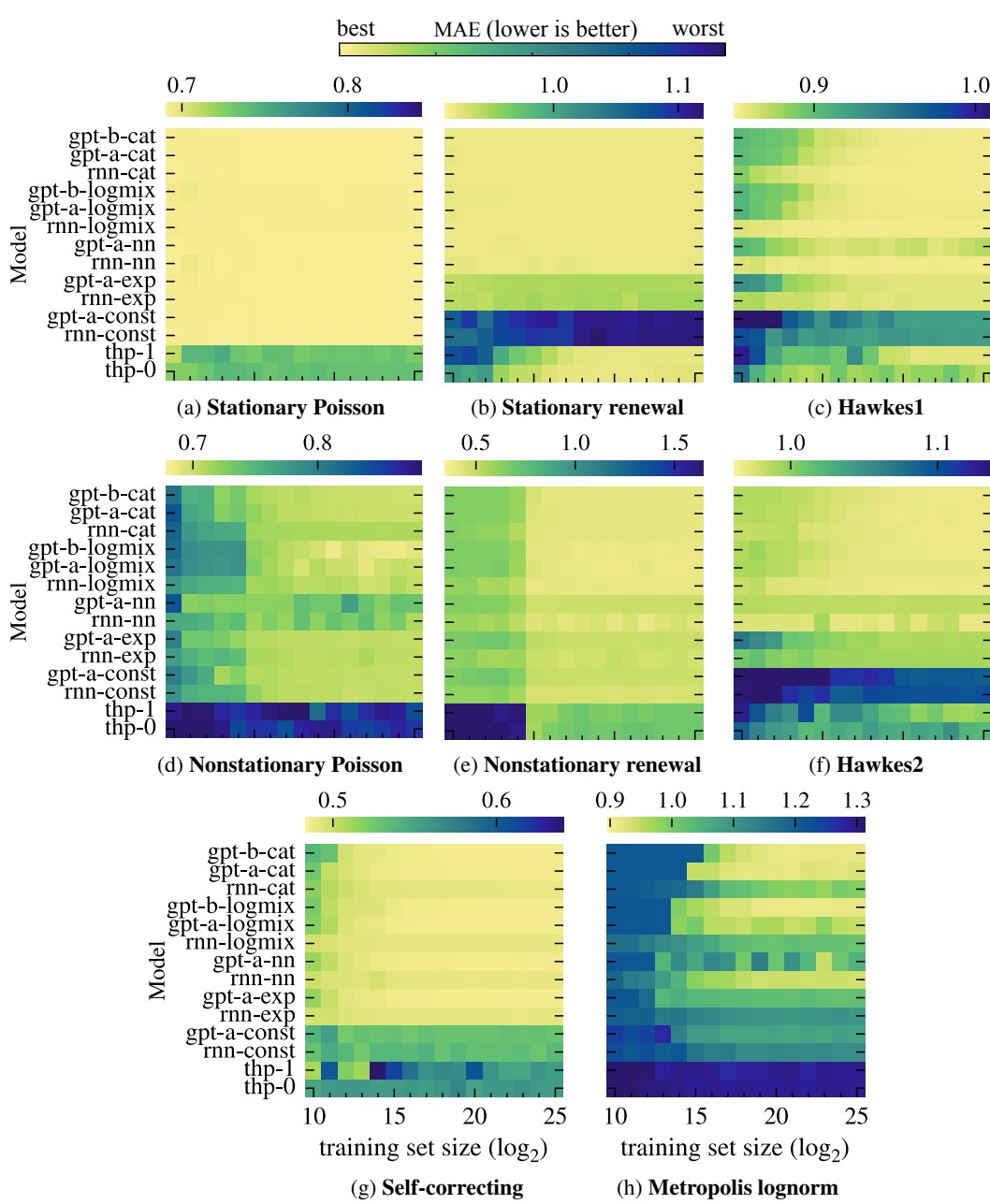

Figure 13: Performance comparison of 14 models in terms of test set MAE across 8 synthetic datasets. There are 16 training set lengths from $2^{10}$ to $2^{25}$. The colormap ranges span each subfigure's full range of values.

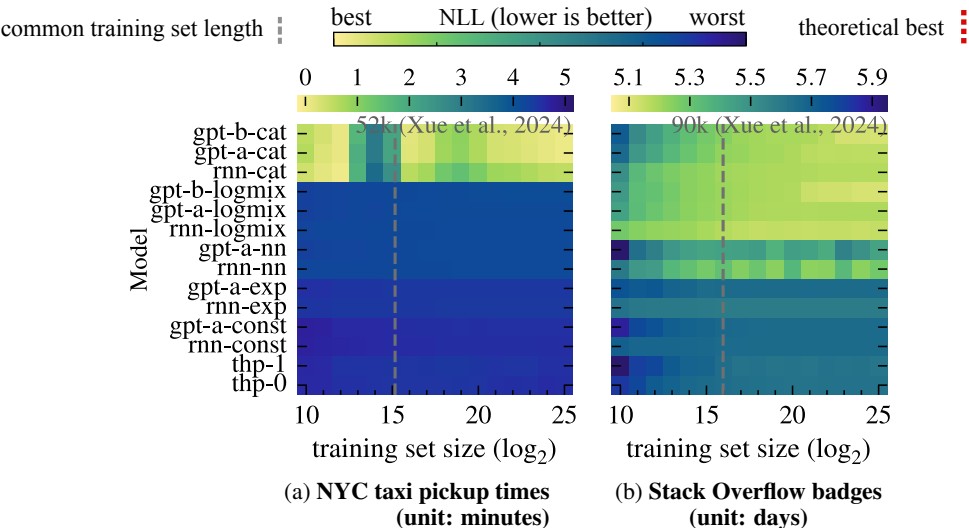

Figure 14: Performance comparison of 14 models in terms of test set NLL across 2 real-world datasets. There are 16 training set lengths from $2^{10}$ to $2^{25}$. The colormap ranges span each sub-figure's full range of values.

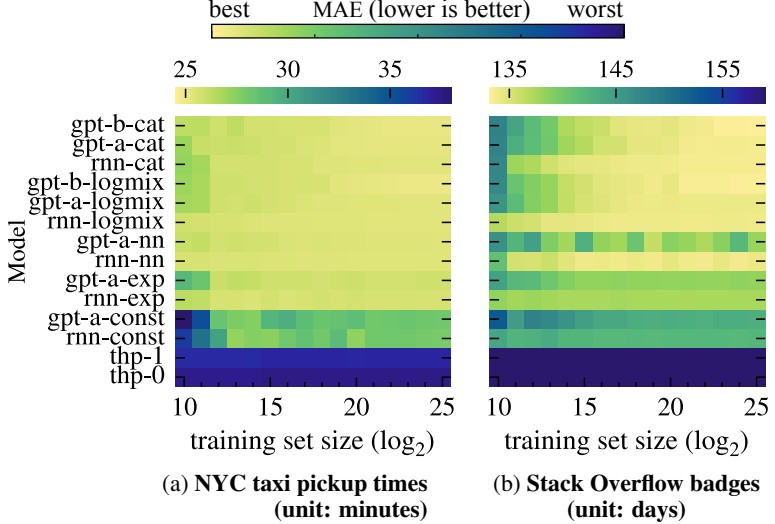

Figure 15: Performance comparison of 14 models in terms of test set MAE across 2 real-world datasets. There are 16 training set lengths from $2^{10}$ to $2^{25}$. The colormap ranges span each sub-figure's full range of values. The inter-event times for the NYC taxi dataset are scaled to minutes, and the Stack Overflow badges dataset is scaled to days.

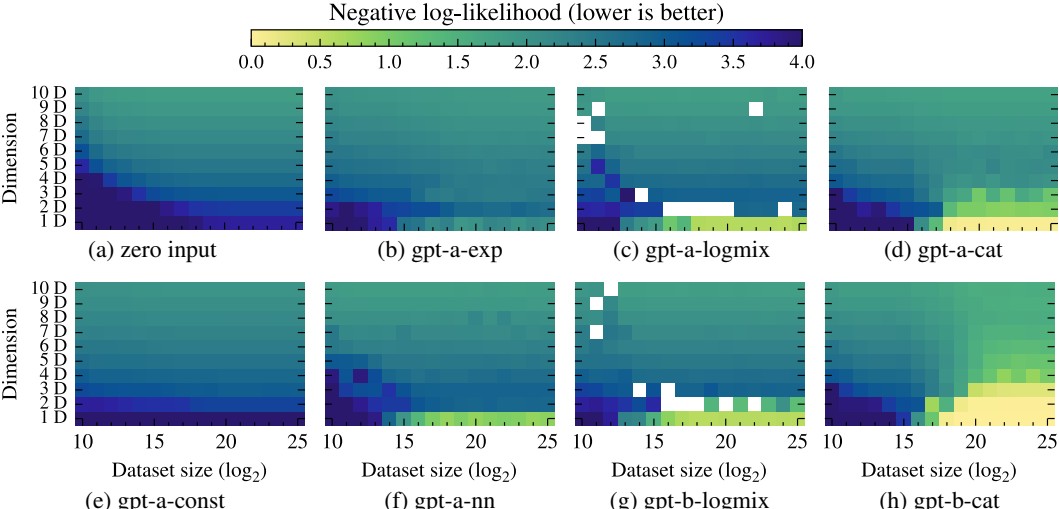

Figure 16: Performance comparison of 8 models in terms of test set NLL. Each model is trained and evaluated over a landscape of 16 training set sizes for 10 modulo datasets. Model (a) takes no input and just outputs the empirical training set distribution; it acts as a baseline. Likelihood is probability mass based and involves integrating over a 1-unit interval for models that output a continuous distribution. White ($\square$) represents a likelihood of 0 ($\infty$ negative log-likelihood) and includes the case where a model assigns a likelihood low enough to encounter numerical issues.

Another observation from the comparison between MAE and NLL is that `thp-0` and `thp-1` score far lower in terms of MAE compared to NLL. These models have two output heads, one for specifying a probability distribution, and another for making point predictions, and the models are trained with a sum of negative log-likelihood and mean-squared error (MSE) loss terms using these heads (Zuo et al., 2020). One possible explanation for the poor MAE scores is that, according to statistical decision theory, the median of a distribution minimizes the expected cost under an MAE cost model, whereas the mean of a distribution minimizes the expected cost under an MSE cost model. That is to say, it is possible that the MSE loss term is suboptimal for the task of point predictions evaluated with MAE.

Infinite negative log-likelihood (zero likelihood) scores on the modulo datasets raise questions of numerical precision, overfitting and test set lengths. The NLL scores for the cyclic datasets (Figure 16) contain zero likelihood scores: the logmix head concentrates sufficiently little probability mass to the interval containing the ground truth event that calculating the log probability produces the floating-point representation of negative infinity (when using `float32` precision). Although this occurs rarely, the relatively large test sets used in this work magnify the chance that this leads to an infinite NLL score. In part, this is a reflection of the issues of NLL as a metric; when viewing the performance of the logmix head in terms of MAE (Figure 11), there is no indication of this issue. A poor way to address this issue is to clamp the output probabilities above a minimum value—this causes the output distribution to no longer be a valid probability distribution by having infinite probability mass on the interval $[0, \infty)$. If the cause is a result of `gpt-a-logmix` or `gpt-b-logmix` being overconfident in certain values, a suitable solution could be to design a loss term that encourages at least one of the mixture components to have a relatively large variance (and non-zero mixture weight).

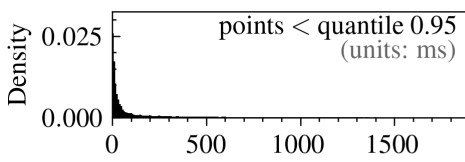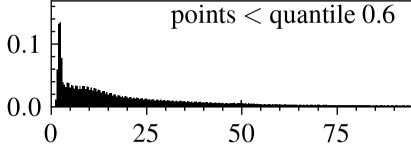

Figure 17: Histograms for the Chicken RGC spike dataset. The histogram extend until the 0.95 quantile (**left**) and 0.6 quantile (**right**). The histograms have 128 bins. The $y$-axis shows normalized counts.

## B DATASETS

This appendix describes the datasets used in Sections 3 to 7. Appendix B.4 and Appendix B.3 explain the extension of existing datasets to accommodate larger training set lengths. Appendix B.5 describes the algorithm behind the Metropolis lognormal dataset and Appendix B.7 gives further details on the modulo datasets.

### B.1 DISTRIBUTION PROPERTIES

Histograms for all real-world datasets are shown in Figure 17, Figure 18 and Figure 19.

#### B.1.1 GINI COEFFICIENT

The Gini coefficients for these datasets are listed in Table 7. The Gini coefficient is a measure of distribution inequality, which is useful in this context for indicating distributions which *might* have a mix of continuous and discrete parts. A distribution where probability is concentrated in a few isolated points in an otherwise large support will have a high Gini coefficient. The NYC Taxi dataset has such a distribution (see Figure 19g). The Gini coefficient is calculated without binning and operates on the resolution of the data type, float32.

Table 7: Gini coefficients for the distribution of inter-event times of the real-world datasets.

| Dataset | Gini coefficient |
|---|---|
| Amazon | 0.000 |
| Yelp airport | 0.148 |
| Yelp Toronto | 0.192 |
| Yelp Mississuaga | 0.361 |
| Twitter | 0.438 |
| Stack Overflow | 0.461 |
| Taobao | 0.587 |
| Last.fm | 0.705 |
| Wikipedia | 0.722 |
| Chicken RGCs | 0.745 |
| PUBG | 0.871 |
| Reddit AskScience | 0.877 |
| MOOC | 0.885 |
| Reddit Politics | 0.906 |
| NYC taxi | 0.974 |

### B.2 SOURCES FOR THE REAL-WORLD DATASETS

The Last.fm, MOOC, Wikipedia and Yelp Toronto datasets used in this work were those made available alongside Shchur et al. (2020). The PUBG, Twitter, Yelp Airport, Yelp Mississauga and two Reddit datasets were those made available alongside Lüdke et al. (2023). The Taobao and Amazon datasets are from Xue et al. (2023). The Taobao and Amazon datasets define fixed train,

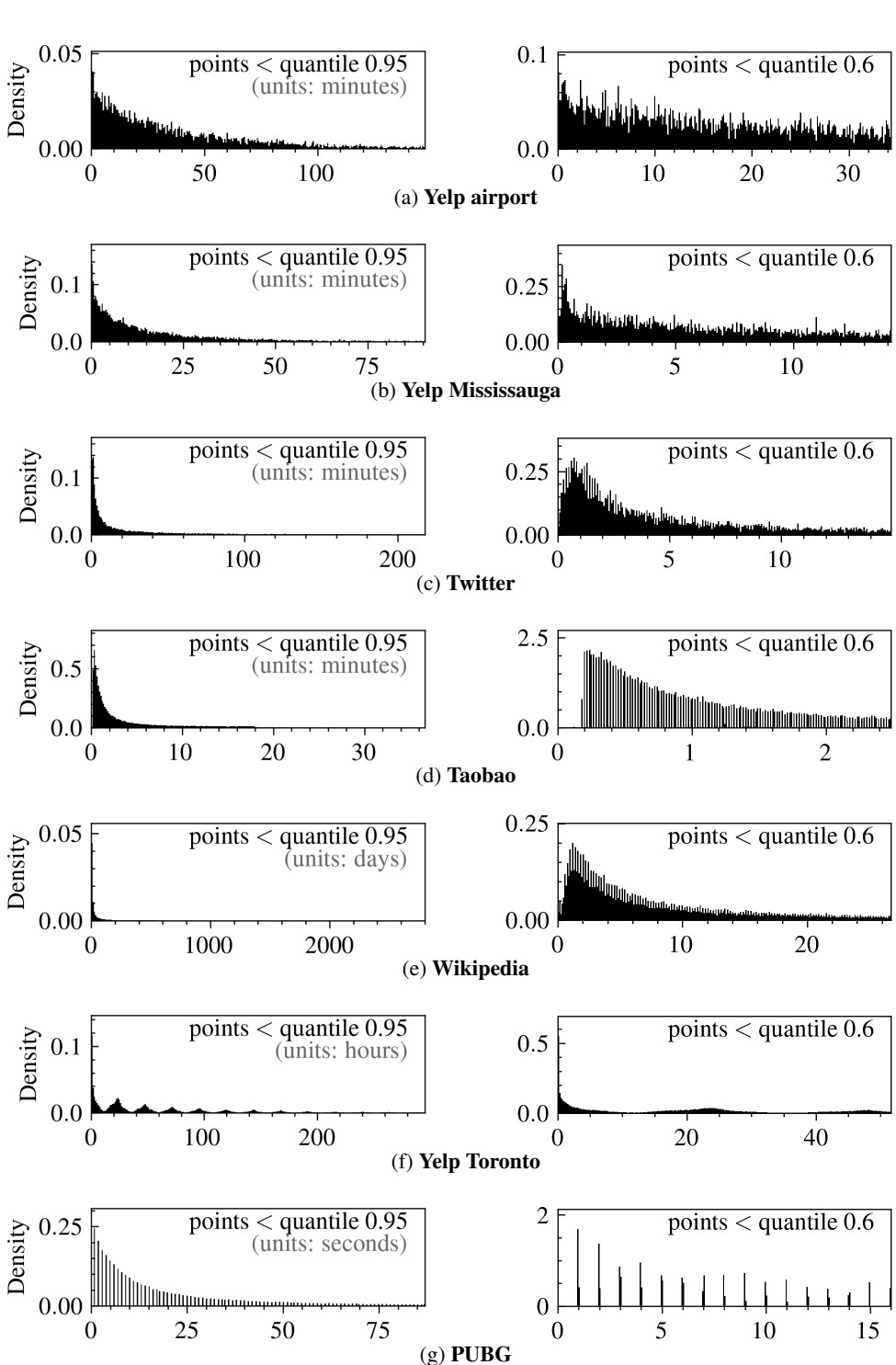

Figure 18: Histograms for 7 real-world datasets. Histograms extend until the 0.95 quantile (**left**) and 0.6 quantile (**right**). All histograms have 128 bins. The $y$-axis shows normalized counts.

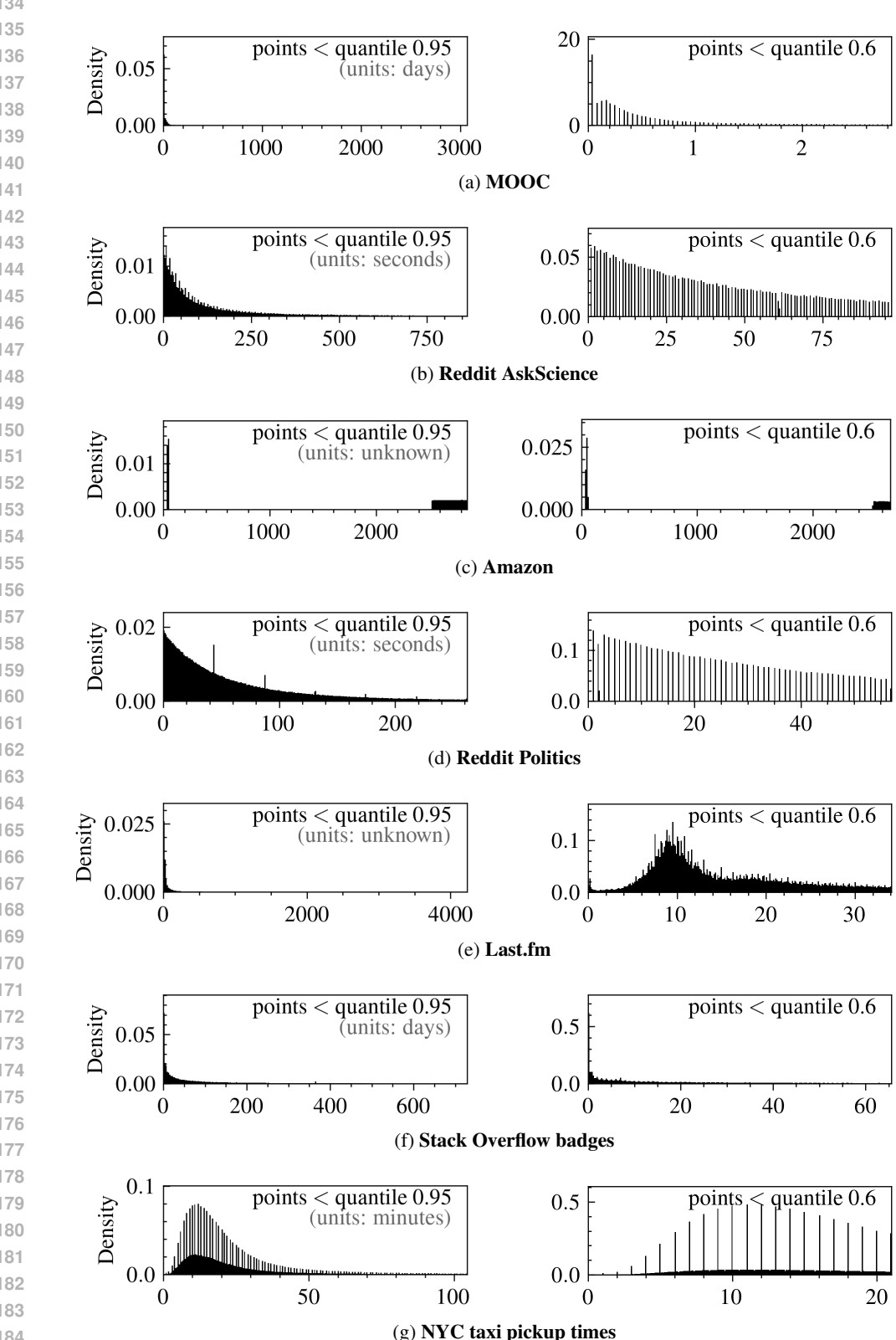

Figure 19: Histograms for 7 real-world datasets. Histograms extend until the 0.95 quantile (**left**) and 0.6 quantile (**right**). All histograms have 128 bins. The $y$-axis shows normalized counts.

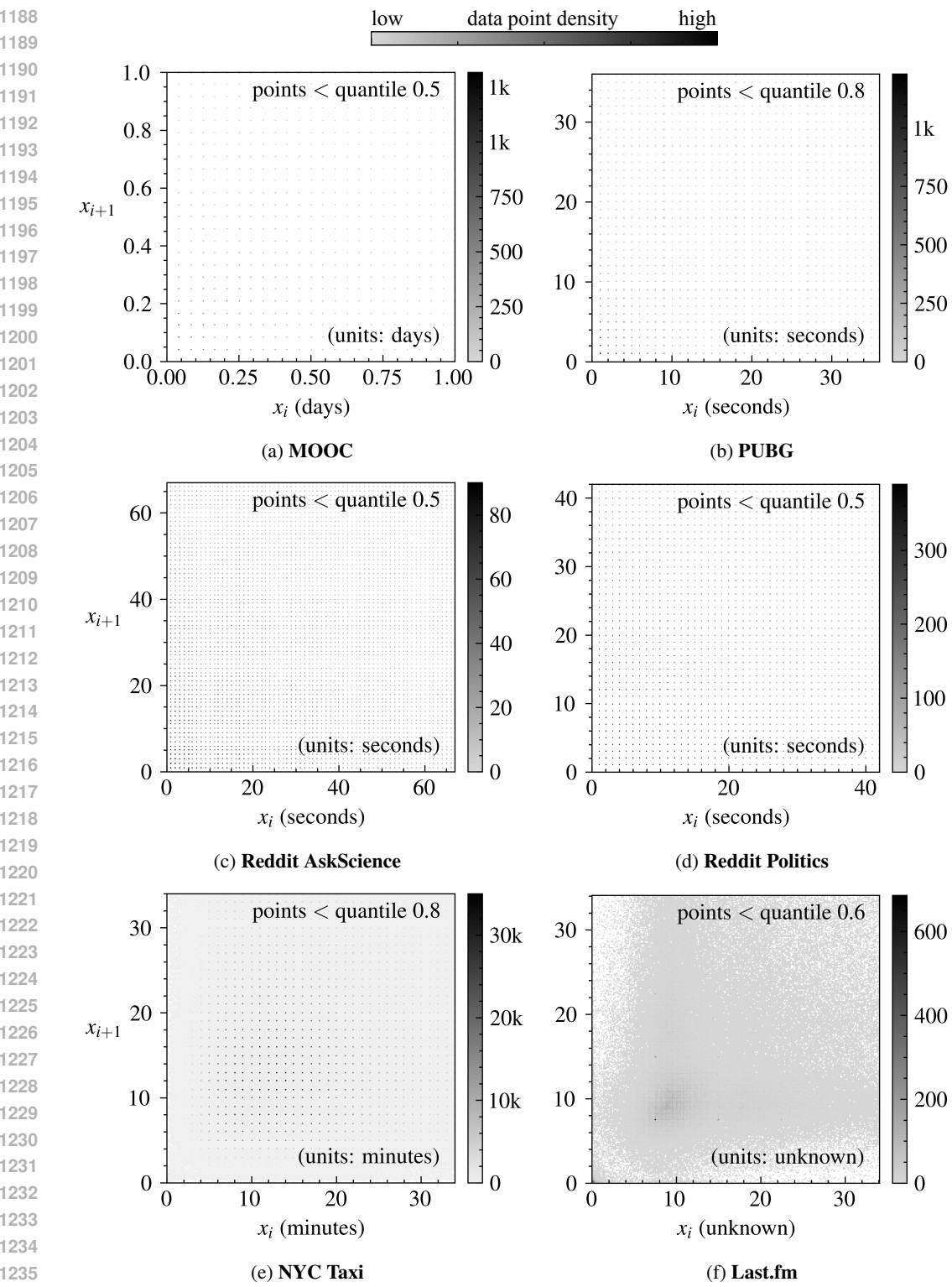

Figure 20: Distribution of pairs of inter-event times, $x_i$ and $x_{i+1}$ for 6 real-world datasets. The first 4 datasets (**MOOC**–**Reddit Politics**) appear to have a grid structure on account of the resolution at which the data was recorded. The last 2 datasets (**NYC Taxi** and **Last.fm**) have a grid structure embedded within a smoother distribution, suggestive of a periodic component to the underlying process. Each figure has a quantile threshold chosen so that the structure is visible. Each figure is a rasterization of event counts into a $256 \times 256$ pixel grid.

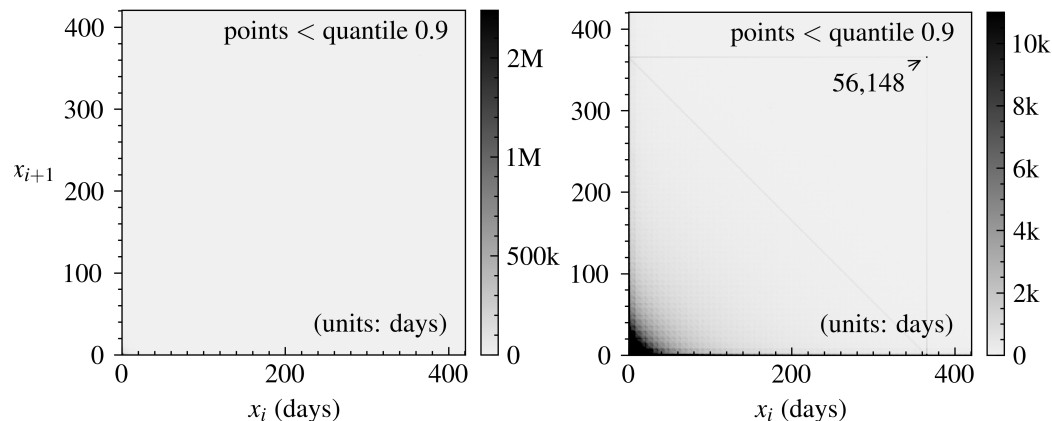

Figure 21: Distribution of pairs of inter-event times, $x_i$ and $x_{i+1}$ for the **Stack Overflow** dataset. The **right** figure is a clipped version of the **left** figure. The peak at ~365 days is small relative to the overall density, but is visible in the **right** figure when the maximum colormap value is reduced. This peak presumably corresponds to a badge typically awarded yearly.

validation and test splits, whereas for the other datasets, a random split into train, validation and test sets was carried out with a ratio of 6:2:2.

### B.3 EXTENSION OF NYC TAXI PICKUP TIMES AND STACK OVERFLOW BADGES

While there are many "real-world" datasets used in literature on TPPs, most are relatively small with less than 1 million events. New York City taxi pickup times and Stack Overflow badges were chosen for this work as the source data contains tens of millions of events, allowing a wide range of training set sizes to be investigated.

Data on taxi pickup times in various cities has been used in numerous works on TPPs. We use the data collected by Whong (2014). This data is used in works such as (Du et al., 2016). The data was obtained by Whong (2014) through filing a Freedom of Information Law Request (FOIL) to the New York City Taxi and Limousine Commission. The data comprises of millions of rows where each row contains (among other information): medallion ID, pickup datetime and dropoff datetime. For example: `["89D227B655E5C82AECF13C3F540D4C", "2013-01-01 15:11:48", "2013-01-01 15:18:10"]`. The pickup and dropoff datetimes are recorded with a resolution of seconds. We process this data by grouping all entries by medallion, sorting by pickup time to create a list of event sequences, and then subtracting adjacent event times to form sequences of inter-event times. Subsets of this list of lists are used to form the training, validation and test sets. The training set is formed by taking the first $n$ sequences such that the $n$ sequences contain at least $2^{25}$ inter-event times. The validation set is taken next followed by the test set such that each has at least $2^{17}$ inter-event times. The remaining events are not used. While data is recorded at a resolution of seconds, we rescale the data to units of minutes. We do no further processing of the data. This is in contrast to Du et al. (2016) who split sequences at gaps larger than 12 hours. We hypothesize that splitting at large gaps would benefit models which are less capable of expressing distributions with peaks far from zero; for example, a model outputting a constant hazard (equivalent to a parameterized Poisson distribution), cannot concentrate probability far from zero without also increasing the distribution's second moment. We argue that there is no fundamental reason why models should not be expected to model these gaps, and so we do not split any sequences.

Timestamps of users receiving badges on the website Stack Overflow were also used by Du et al. (2016). In order to complete training, validation and test sets of size $2^{25}$, $2^{17}$ and $2^{17}$, we obtained a more recent data export (last updated 29th August 2024) of the data from the Stack Overflow website (Overflow, 2024). Timestamps for this dataset are recorded with a resolution of milliseconds. Processing was carried out in the same manner as for the NYC taxi dataset described above, this time grouping by the column `UserId`, and rescaling to units of days. At the time of download, Stack

Overflow stipulated that the data was provided under the condition that the data is for the users' own use; and so we do not distribute a copy.

## B.4 SYNTHETIC DATASETS FROM (OMI ET AL., 2019)

We generate the 7 synthetic datasets from (Omi et al., 2019) following the released code that accompanied that work. We extended the length of the generated sequences to fill training, validation and test sets of size $2^{25}$, $2^{17}$ and $2^{17}$ respectively.

## B.5 METROPOLIS LOGNORMAL PROCESS

The Metropolis lognormal process is an intentionally poor sampler of lognormal distribution, implemented with a modified Metropolis-Hastings algorithm, described in Appendix B.6 and Algorithm 1. The sampler's target distribution is a standard normal distribution, and the events of the process are obtained by taking the log of outputs of the sampler. A sampler uses a stochastic transition matrix like that shown in Figure 23. The proposed samples are generated with another modified Metropolis-Hastings sampler. In total, there is a stack of 3 such samplers.

### B.5.1 MOTIVATION

The Metropolis lognormal process was designed to be decodable to a high degree given sufficient event history and compute. It was also designed so that unconditional inter-event times follow a relatively simple distribution—this is intended to match real-world processes which can be fully deterministic given enough information yet have aggregate properties that follow simple distributions. The Metropolis lognormal process achieves both of these properties by being a poor sampler of a lognormal distribution. The inter-event times are approximate draws from a lognormal distribution, achieving one of the goals above, but the sampling is determined by fixed transition matrices, which can be decoded given a sufficiently long sequence.

## B.6 SEQUENCE GENERATION

The sequence generation algorithm is shown in Algorithm 1. This section gives an overview of the algorithm.

A sequence is generated as follows. First, a transition matrix defining a cycle between states is fixed. We use 20 states. The domain, $(-\infty, \infty)$, is split into 22 intervals; 2 outer intervals, $(-\infty, 4.3)$ and $(4.3, \infty)$, which each contain a small amount $(10^{-5})$ of the probability mass of a normal distribution, and 20 intervals that share the remaining (>99%) probability mass evenly. The 20 states correspond to the 20 inner intervals, while the 2 outer intervals are not used. The Metropolis-Hastings sampling process uses a proposal distribution located at the previous sample. We modify this process to first follow the transition matrix to a new state before sampling with a proposal distribution from the corresponding location in the new interval. We nest this process by having the proposal distribution also be a pseudo-normal distribution generated by the same process but with a separate transition matrix. The log of the generated values become a sequence of inter-event times.

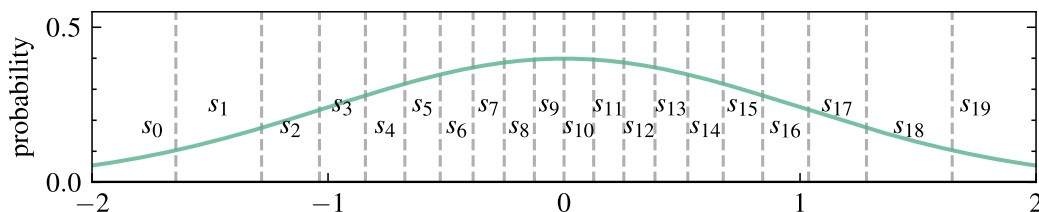

Figure 22: The interval $(-4.26, 4.26)$, which contains all but $2 \times 10^{-5}$ of the probability mass of the standard normal distribution, is divided into 20 subintervals, each containing equal probability mass with respect to the standard normal. The outer intervals, $(-\infty, -4.26)$ and $(4.26, \infty)$, are not used. The intervals are labelled $s_0, s_1, ..., s_{19}$ corresponding to the states used in the Metropolis-Hastings sampler. Here, the figure zooms in to $(-2, 2)$ to see the shorter intervals more clearly.

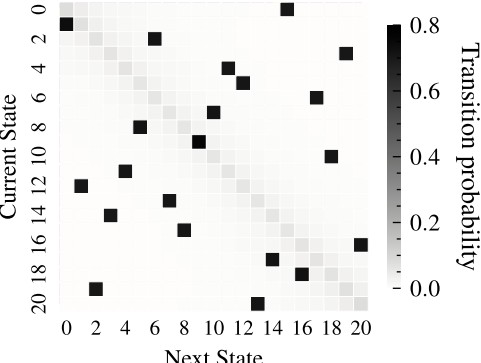

Figure 23: The stochastic state transition matrix used by the outermost sampler. The preponderance of probability is assigned to the next state of a random cycle, with a small amount of probability assigned down the diagonal. Reducing the probability assigned to the diagonal increases the determinism of the sampler, but reduces how well the sampler approximates the target distribution.

**Algorithm 1** Modified Metropolis-Hastings sampler. A subset of the real line $(t_{\min}, t_{\max})$ is subdivided into $S$ intervals of equal probability mass according to the target distribution $\pi$; these $S$ intervals are the states. Before querying the proposal distribution, the algorithm transitions according to a stochastic transition matrix between these states. To calculate the forward and reversal probability, an approximating distribution, `innerPdf`, is used. The sampler calls the step function in a loop, and, unlike standard Metropolis-Hastings, repeated samples are discarded. The samplers can be stacked; in this work, we stack 3 such samplers, and for the base distribution $q_0$, a deterministic cycle between 30 quantile points of a normal distribution is used. This makes the sequence deterministic up to the randomness implied by the transition matrices, $T_1, T_2$ and $T_3$ and the uniform sampling, $\mathcal{U}$.

1: **function** mapPos$(x, s_{\text{from}}, s_{\text{to}})$
2:     Compute the relative position of $x$ within the interval of state $s_{\text{from}}$ and map it to the corresponding position in the interval of state $s_{\text{to}}$.
3:         **return** mapped position
4: **function** posToState$(x)$
5:     Determine the state the given position $x$ is in.
6:         **return** state
7: **function** Sampler.**init**$(x_0, \pi, T, \text{innerSampler}, \text{innerPdf})$
8:     The innerSampler is called to generate new proposals, and innerPdf is called to approximate the distribution of the innerSampler for the purpose of calculating the reversal probability.
9:         **State:** $x_t \leftarrow x_0$
10:         **State:** $\pi, q, T,$ innerSampler, innerPdf
11: **function** Sampler.**step**
12:         $s_t \leftarrow$ posToState$(x_t)$                                      ▷ Determine current state
13:         $s' \leftarrow$ Sample from $T(s' \mid s_t)$                               ▷ Transition to next state
14:         $x' \leftarrow$ mapPos$(x_t, s_t, s')$
15:         $x_{t+1} \leftarrow$ innerSampler$(x_{t+1} \mid x')$                          ▷ Propose new sample
16:         $p_{\text{forward}} \leftarrow T(s' \mid s_t)\,\text{innerPdf}(x_{t+1} \mid x')$
17:         $p_{\text{reverse}} \leftarrow \sum_{i \in S} \text{innerPdf}(x_t \mid \text{mapPos}(x_{t+1}, s', s_i))\, T(s' \mid s_i)$
18:         Compute acceptance ratio:
$$\alpha \leftarrow \frac{\pi(x_{t+1})\, p_{\text{reverse}}}{\pi(x_t)\, p_{\text{forward}}}$$
19:         $u \leftarrow$ Sample from $\mathcal{U}(0, 1)$
20:         **if** $u < \alpha$ **then**
21:             **return** $x_{t+1}$                                          ▷ Accept proposal
22:         **else**
23:             **return** $x_t$                                      ▷ Reject, keep current state
24: **function** Sampler.**sample**
25:         $x_{t+1} \leftarrow$ Sampler.step
26:         **while** $x_{t+1} = x_t$ **do**                                   ▷ Ignore repeated samples
27:             $x_{t+1} \leftarrow$ Sampler.step
28:         $x_t \leftarrow x_{t+1}$                                          ▷ Update state
29:         **return** $x_{t+1}$
30: **function** nestedSampler$(q_0)$
31:         $\pi_1 \leftarrow \mathcal{N}(0, 1)$
32:         $\pi_2 \leftarrow \mathcal{N}(0, \frac{1}{3})$                              ▷ Inner samplers are narrower
33:         $x_0 \leftarrow 0$
34:         $T_1, T_2, T_3 \leftarrow$ sample transition matrices
35:         $q_1 \leftarrow$ Sampler.init$(x_0, \pi_1, T_1, q_0, \pi_2)$
36:         $q_2 \leftarrow$ Sampler.init$(x_0, \pi_2, T_2, q_1, \pi_2)$
37:         $q_3 \leftarrow$ Sampler.init$(x_0, \pi_2, T_3, q_2, \pi_2)$
38:         **return** $q_3$

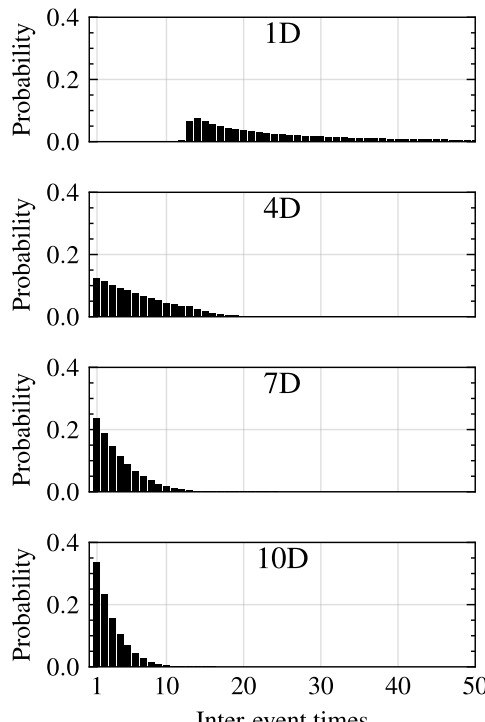

Figure 24: Training sets. Distribution of inter-event times for overflow datasets of dimension 1, 4, 7 and 10 (from top to bottom). The distribution is the count distribution of inter-event times over the $0.8 \times 2^{17}$ sequences in the training set.

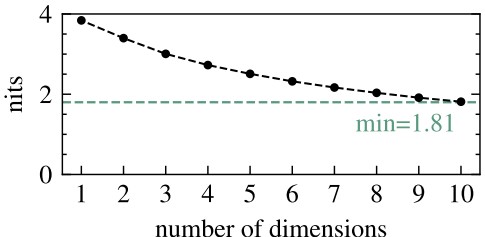

Figure 25: Entropy of the count distribution of inter-event times for the 10 training sets.

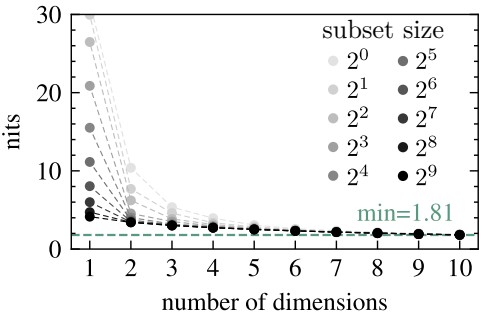

Figure 26: Mean cross-entropy between the distribution of inter-event times in the whole training set and in subsets of the training set. For 10 subset sizes ranging from $2^0$ to $2^9$, the mean cross-entropy is calculated using 100 random subsets of that size. Subset sizes $2^9$ and above are indistinguishable on the figure.

### B.7 MODULO SEQUENCES

This section describes further details of the modulo "overflow" sequences. Appendix B.7.1 describes the properties of the inter-event time distributions of the modulo sequences. Appendix B.7.2 rephrases the sequences in terms of cyclic groups, which allows the easy deduction of the period of the sequences. Appendix B.7.3 describes analogous continuous-time sequences. Appendix B.7.4 explains the motivation behind the bounds on the velocity $v$ used to generate the sequences in this work.

### B.7.1 PROPERTIES OF THE MODULO DATASETS

This section builds some intuition on the modulo datasets. Figure 24 shows the distribution of inter-event times for the 1, 4, 7 and 10-dimension datasets. With increasing dimension, the distribution of inter-event times shifts to concentrate near 1, with a roughly exponential shape. The entropy of all 10 datasets is shown in Figure 25. The decreasing entropy with increasing dimension explains how the NLL scores for the zero input model (Figure 16a) decrease (improve) with increasing dimension. This is also demonstrated in Figure 26, where the cross-entropy between the full training set and smaller training sets predicts the NLL scores of the zero input model; the distributions of subsets with more than $2^9$ sequences ($2^{19}$ events) can be seen to be very similar to the overall distribution. Finally, Figure 27 uses $\mathcal{V}$-information (Xu et al., 2020) as applied by Ethayarajh et al. (2022) to gauge how difficult the datasets are for the gpt-b-cat model to learn beyond what the zero input model can learn. Interestingly, for some of the dimensions, there is a U-shaped relationship where at both small and large dataset sizes, the gpt-b-cat model can outperform the zero input model to a greater degree compared to intermediate dataset sizes.

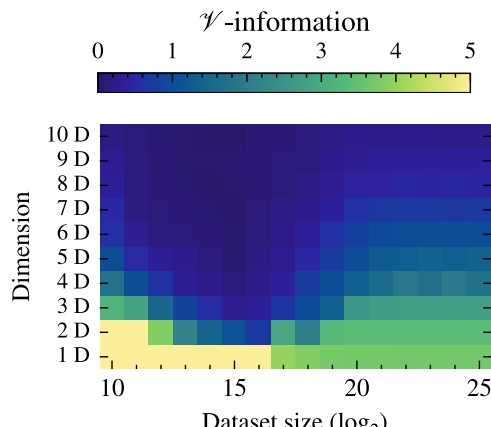

Figure 27: $\mathcal{V}$-information calculated following Ethayarajh et al. (2022) as $H_\mathcal{V}(Y) - H_\mathcal{V}(Y|X)$, where $H_\mathcal{V}(Y)$ is estimated from the cross-entropy from Figure 26 and the `gpt-b-cat` model results in Figure 16 are used for $H_\mathcal{V}(Y|X)$. The set $\mathcal{V}$ is the set of all possible mappings describable by the `gpt-b-cat` architecture. Estimating $H_\mathcal{V}(Y)$ from the cross-entropy is possible as a `gpt-b-cat` model trained with no input will match the empirical distribution of the inter-event times in the provided training set.

### B.7.2 CYCLIC GROUP PERSPECTIVE

The sequence $(\mathbf{a}_k)_{k=0}^\infty$ of particle positions in $d$ dimensions can be understood as stepping through a sub-cycle of a cyclic group. In 1-dimension, for a given $n \in \mathbb{N}$, two elements $c$ and $g$ of the cyclic group $\mathbb{Z}/n\mathbb{Z}$ define the sequence $(a_k)_{k=0}^\infty$ as $cg^k$. The overflow sequence contains every $k$ such that $cg^k < cg^{k-1}$. Moving to higher dimensions, let $G$ be the product of $d$ cyclic groups $G_1, G_1, ...G_d$ of order $n_1, n_2, ...n_d$. Let $c$ and $g$ be elements of $G$ and, as before, define the sequence $(g_k)_{k=0}^\infty$ as $cg^k$, with the overflows now defined as the $k$s where any element of $cg^k$ is less than the previous element in $cg^{k-1}$. If all $n_1, n_2, ..., n_d$ are chosen to be prime, then it is guaranteed that all sequences will have a period of $n_1 \times n_2 \times ... \times n_d$. As long as this product is less than the input length of a model, we can be confident that a model is not simply repeating a loop seen in its input.

### B.7.3 CONTINUOUS SEQUENCES

A continuous analogue in 1-dimension is to consider a particle starting from position $p_0 \in \mathbb{R}$ and moving in a straight line with velocity $v \in \mathbb{R}$. An event occurs at every time $t > 0$ such that $p_0 + vt = kn$ for some $k \in \mathbb{N}$ and fixed $n \in \mathbb{R}$. An event sequence is then defined as the ordered set of all such $t$. This extends naturally to higher dimensions by considering vectors $\mathbf{p_0}$, $\mathbf{v}$ and $\mathbf{n}$, and having events occur every time one of the components of $\mathbf{p_0} + \mathbf{v}t$ is an integer multiple of the corresponding component of $\mathbf{n}$.

More complicated sequences can be generated by adding more particles or making them bounce rather than pass through faces, or making the task to determine the next collision time of a set of particles with mass moving in a box.

### B.7.4 BOUNDS ON V

An upper bound on $v$ prevents the inter-event time distribution from collapsing to $\mathbb{P}(\Delta t = 1) = 1$ as the dimension increases. For example, in 10-dimensions, if components of $\mathbf{v}$ are around 500, with the grid being defined by $\mathbf{n} = [1021]^d$, then each component overflows every 2-3 steps, resulting in almost every step containing an event. In this work, the upper bound on $v$ was set to $80$ to limit this effect.

The reason for the lower bound of 10 is that it sets the maximum inter-event time at around 100 (103). The benefit of this is that it allows the discrete approach to encode all possible event times with a reasonable array size. This marks a similarity to the spike prediction task, where 81 bins were sufficient to encode all events of interest. If there is a larger maximum inter-event time, then

the discrete model can use a larger output array or group events into a single output bin, similar to how bins were chosen in Section 3.

## C    MODEL DETAILS

This section covers some details on implementing the models used in this work. More details on how a categorical distribution can be mapped to a distribution over $[0, \infty)$ are covered in Appendix C.1. Encoding event times to be input to `gpt-a` and `gpt-b` is covered in Appendix C.3. Appendix C.4 describes the implementation of the existing models used in this work.

### C.1    BIN WIDTHS AND DENSITIES FOR THE CATEGORICAL HEAD WITH CONTINUOUS DATA

When the target distribution is constrained to a small finite number of possibilities, as is the case for the spike prediction task and the modulo datasets, the categorical head can output an array of unnormalized probabilities with one element per possible event. This strategy does not work when the target distribution is closer to being continuous, as is the case for the datasets used in Sections 3 to 4. As described in Section 3, in this work, we follow Kvamme & Borgan (2021) and split the domain of the target distribution into intervals (as many intervals as we have output elements) that evenly share the probability mass of the training set's empirical distribution. In this work, the domain is always assumed to be $[0, \infty)$, and the categorical models always output 128 elements. The endpoints of the intervals are determined as follows. The first interval begins at 0 and the last interval extends to $\infty$. The 127 remaining boundaries are assigned to the $1/128, 2/128, ..., 127/128$ quantiles of the inter-event times from the training set. The quantiles are calculated using the default `numpy` quantile function, which uses the linear interpolation method from (Hyndman & Fan, 1996). Figure 28 shows an example output from a trained `gpt-a-cat` model with 128 output elements. The probability density within each interval is $\frac{c_i}{w_i}$, where $c_i$ is the probability mass assigned to the $i$-th interval and $w_i$ is the width of the $i$-th interval. Thus, for all bins except the final bin, the output of the categorical head is interpreted as a piecewise constant density function.

For the final infinite length interval, the probability density is given by: $f(t) = c_{128}\lambda e^{-\lambda t}$, where $c_{128}$ is the probability mass assigned to the final interval, and $\lambda$, the rate parameter, is chosen to be the length of the 2nd last interval. In practical applications where predictions are updated when new information becomes available, the final bin may not require a density function, and the probability mass assigned to it is sufficient. A density for the last bin is used in this work so as to enable comparison to other models using the density-based NLL metric.

When calculating intervals, if a single inter-event time is frequent enough to represent more than $\frac{1}{N}$ of the probability mass, then a zero width interval could be encountered, leading to an infinite probability density. To prevent this singularity, we impose a minimum interval width of $2^{-17}$ to avoid numerical issues. While not explored in this work, the choice of minimum can act as a form of regularization, with wider minimums providing stronger regularization.

### C.2    GPT-A AND GPT-B

The `gpt-a` and `gpt-b` transformers follow the standard GPT-2 architecture ((Radford et al., 2019)) with the layer configurations (layers-heads-embedding size) set to 2-4-16 and 6-4-32 respectively. This gives `gpt-a` and `gpt-b` 108k and 1.20M trainable parameters, respectively. The standard GPT-2 small configuration defined by Radford et al. (2019) uses a 12-12-768 configuration, highlighting that both `gpt-a` and `gpt-b` are very small in the context of GPTs used as large language models. Computational limits prevented the investigation of larger models.

### C.3    INPUT ENCODING FOR GPT BASED MODELS

When the `gpt-a` and `gpt-b` architectures are used for the spike prediction task, the input to these modules is a sequence outputted by the CNN encoder. When the `gpt-a` and `gpt-b` architectures are used on their own, for example in Section 3, the scalar values representing inter-event times are converted to a vector representation according to Algorithm 2 before being input into the transformers. This approach is similar to that used by Zuo et al. (2020). We do not hard-code the sinusoidal

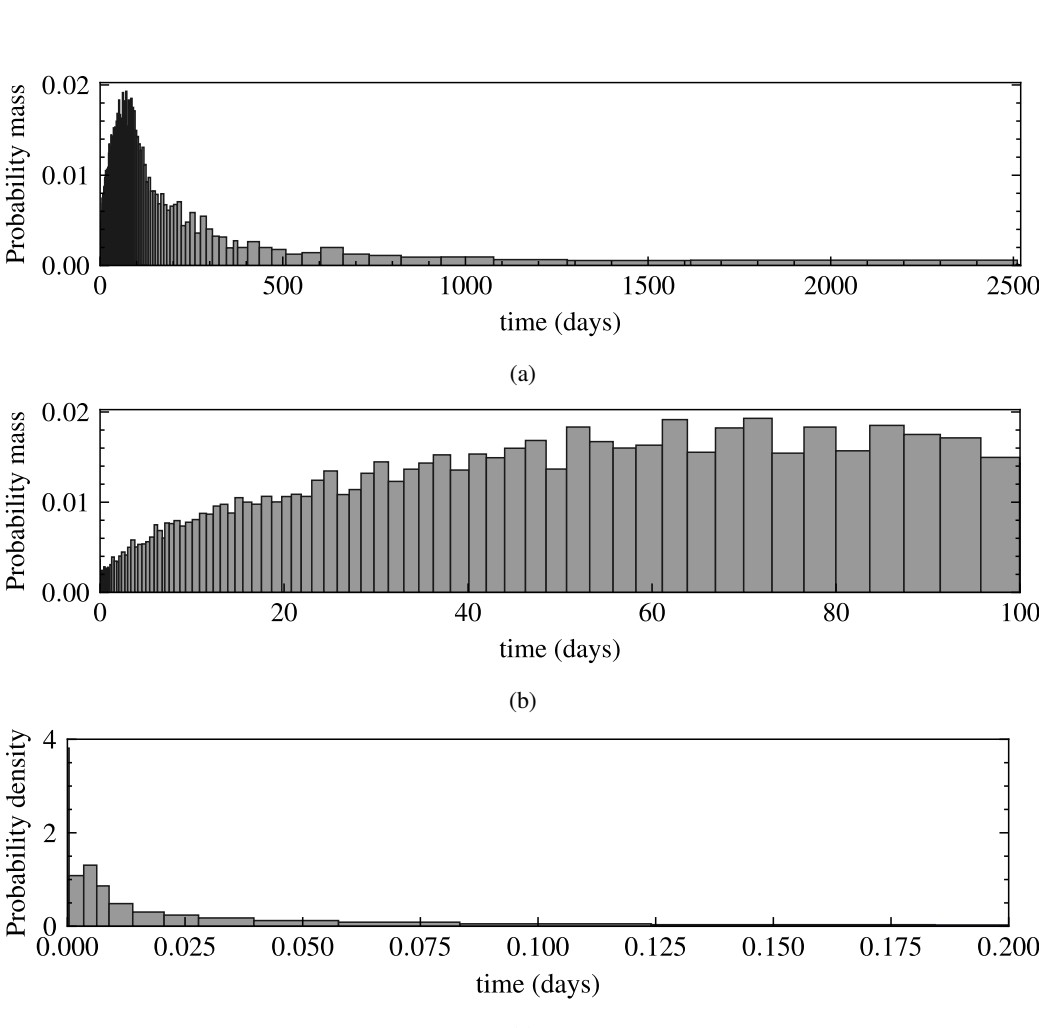

Figure 28: An example output of a `gpt-a-cat` model trained on the $2^{25}$ length Stack Overflow badges training set. **(a):** the probability mass assigned to each of the 128 output intervals. The bar widths correspond to the interval widths and heights correspond to the probability mass assigned to the intervals by the model. The final infinite interval starts from 2511 and is assigned only a small probability ($7.5 \times 10^{-5}$), so cannot be seen. **(b):** the same output as (a), but zoomed in to the interval $(0, 100)$. **(c):** zoomed into the interval $(0, 0.2)$, but this time, the bar heights correspond to the probability density over the intervals (mass/width). In this example, the probability density is highest for the very narrow intervals near 0.

frequencies but choose appropriate scales so that both long and short intervals seen in the data are captured.

---

**Algorithm 2** Encode a scalar value $v$ into a vector of even size $n$. $v_{\text{epsilon}}$ should be chosen near the smallest inter-event time expected to be encountered and $v_{\text{max}}$ should be chosen to be near the largest time expected to be spanned by inputs to the model, and will depend on the length of a model's input. In this work, we choose $v_{\text{epsilon}}$ as the smallest inter-event time observed in the training set, and $v_{\text{max}}$ as the largest time between events $t_i$ and $t_{i+L}$ where $L$ is the input length of the model.

---

1: **function** encodeValue($v, v_{\text{epsilon}}, v_{\text{max}}, n$)
2:      $\text{scale}_{\min} \leftarrow 2\pi/v_{\text{max}}$                            ▷ Or use $\frac{3}{2}\pi$ to add a little more range
3:      $\text{scale}_{\max} \leftarrow 2\pi/v_{\text{epsilon}}$
4:      $\text{scale} \leftarrow \text{linspace}(\log(\text{scale}_{\min}), \log(\text{scale}_{\max}), n/2)$          ▷ A vector of length $n/2$
5:      $x_1 \leftarrow \cos(\exp(\text{scale}) \cdot v)$
6:      $x_2 \leftarrow \sin(\exp(\text{scale}) \cdot v)$
7:      $x \leftarrow \text{concat}(x_1, x_2)$
8:      **return** $x$

---

### C.4 IMPLEMENTATION OF EXISTING MODELS

The `const`, `exp` and `nn` heads are implemented following the code accompanying (Omi et al., 2019). The `logmix` head is implemented following the code accompanying Shchur et al. (2020). The `thp-0` and `thp-1` models are implemented partially following (Zuo et al., 2020), but deferring to the implementation accompanying (Xue et al., 2023) for a number of bug fixes to the original model.

One deviation from these implementations is in model initialization. In order to improve convergence speed and stability while training, all models were initialized so that their outputs have first and second moments similar to those of the training set's empirical distribution.

### C.5 LIMITATIONS OF EXISTING MODELS

The existing output structures considered in this work have limitations in terms of distribution flexibility and training stability, both of which contribute to preventing them from representing multimodal distributions with very narrow peaks.

#### C.5.1 DISTRIBUTION FLEXIBILITY

The constant intensity, exponential intensity and the softplus intensity (used by `thp-0` and `thp-1`) correspond to distributions over $[0, \infty)$ with 1 or 2 parameters. With only 1 or 2 parameters, they do not have the flexibility to represent distributions where there are multiple separated peaks of high probability density, which is a property of several of the real-world datasets (see Figure 20).

**The constant intensity** output corresponds to the exponential distribution. The exponential distribution has a single scalar parameter with the distribution's mode fixed at 0. The distribution cannot concentrate probability at any non-zero point.

**The exponential intensity** output corresponds to the Gompertz distribution. The Gompertz distribution has two parameters, a scale and a shape parameter. It is a unimodal distribution so cannot concentrate probability at two or more separated points.

**The softplus intensity** function corresponds to a non-standard unimodal probability distribution with two parameters. Its unimodal shape is inferred from the monotonicity of the softplus function. Being unimodal, it is restricted in its ability to concentrate probability at multiple separated points.

#### C.5.2 MIXTURE OF LOGNORMALS INSTABILITY

A mixture of lognormals can theoretically represent distributions with multiple sharp peaks. However, when training models with this output structure on datasets where there is a significant discrete component to the inter-event distributions, we do not observe the models being able to strongly

concentrate probability in narrow peaks. We argue that such models struggle to represent such distributions on account of training instability in the presence of singularities.

A mixture of lognormals on $t \in (0, \infty)$ is equivalent to a mixture of Gaussians on $y \in (-\infty, \infty)$, where $y = \log(t)$, and so models that output a mixture of lognormals can be thought of as outputting a mixture of Gaussians on a log scale. A mixture of Gaussians can produce any number of modes, up to the number of components in the mixture.

The process of fitting these mixtures using the expectation-maximization (EM) algorithm demonstrates how this flexibility does not necessarily allow the mixture to represent distributions with sharp peaks. When fitting a mixture of Gaussians using the EM algorithm, a component's variance can collapse toward zero at a single point. To avoid this, strategies such as introducing priors on the parameters or resetting the EM optimization procedure are used Bishop (2006).

When using stochastic gradient descent to train a neural network with a mixture of lognormals output, we argue that encountering such singularities can manifest as unstable training and prevent models from expressing distributions with sharp peaks. A simple case study (Figure 29) shows gpt-a, outputting a 2-component lognormal mixture, trained on a dataset generated from a mixture of a lognormal distribution and a point mass. As one of the output components narrows around the single point, sudden spikes in gradient cause the parameters of the component to be perturbed and its mixing weight to suddenly decrease. "Real-world" datasets such as the NYC taxi dataset have such point masses, and training stability in the presence of these sharp distributions is a candidate explanation for why the logmix head is not observed to concentrate probability at such points in a stable manner. To prevent the instability, regularization could be introduced in the form of a prior on the variance of each component. Doing so amounts to specifying a resolution of interest for the task and would be further evidence of the benefit of acknowledging the discrete nature of recorded event times.

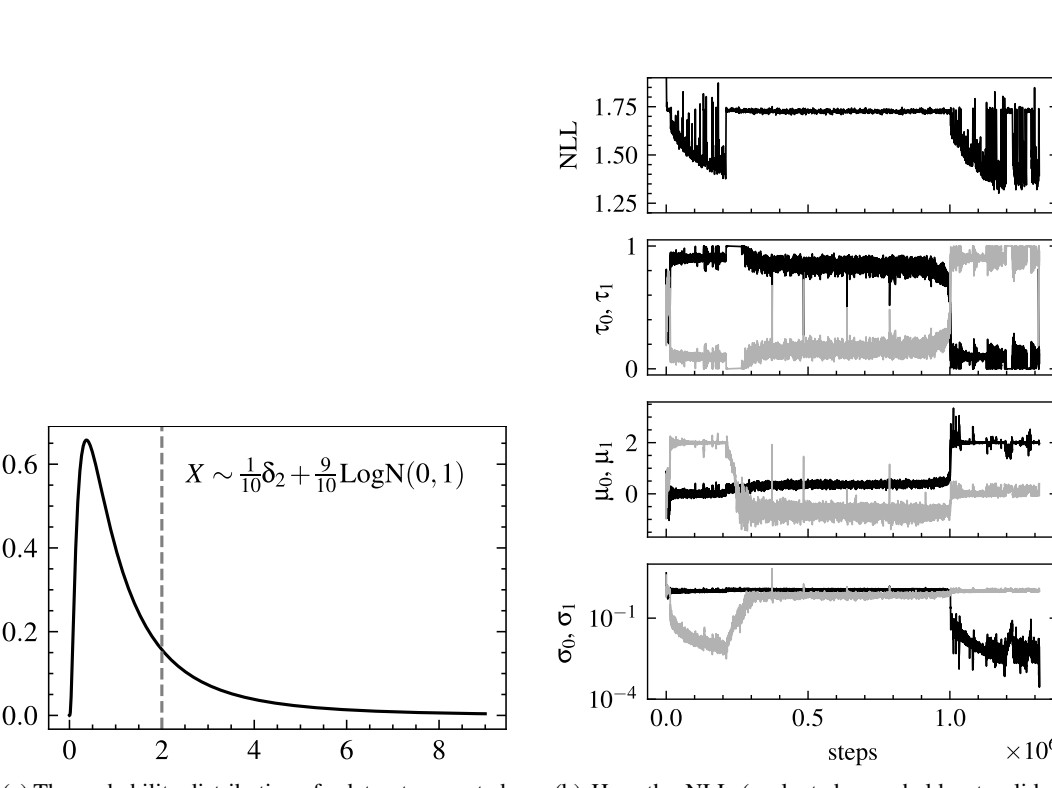

(a) The probability distribution of a dataset generated from a mixture of a point mass at $t = 2$ and a lognormal distribution with parameters $\mu = 0$ and $\sigma = 1.0$.

(b) How the NLL (evaluated on a held-out validation set) and the model's six parameters change while training.

Figure 29: The instability of a logmix output head when a dataset has a probability mass at a certain point. **(a):** shows the distribution of inter-event times. The dataset is formed from a mixture where 10% of values come from a point source at $t = 2$, and the remaining values are drawn from a lognormal distribution with parameters $\mu = 0$ and $\sigma = 1.0$. All events are independent. The model used is a `gpt-a` stem with a 2-component lognormal mixture output. The model is trained using stochastic gradient descent with momentum (0.9) and a learning rate of $5 \times 10^{-4}$. **(b):** shows the mixture parameters (the model output) and the NLL (calculated on an evaluation set) over the course of training. The six model parameters are the mixing parameters ($\tau_0, \tau_1$) and the means and variances ($\mu_0, \mu_1, \sigma_0, \sigma_1$) of the two underlying Gaussian distributions. Notably, neither $\mu_0$ nor $\mu_1$ approach 2.0 in a stable manner. Training stopped when evaluating the distribution median produced `NaN` values. This instability occurs reliably across runs.

# D  SPIKE PREDICTION

This appendix is supplementary to Section 6: *A categorical output is effective for spike prediction*.

## D.1  DATASET

The spike prediction task uses a dataset constructed from 16 recordings of chicken RGC spike activity; the recordings were carried out by Seifert et al. (2023) using a multi-electrode array. Each recording is 15 minutes long and contains activity of multiple RGCs. The processing of the recordings to produce a dataset follows the procedure described by Doran et al. (2024), who created a dataset from 1 of the 16 recordings. In total, across the 16 recordings, 1611 cells are used.

Both the stimulus and spike times are stored at a sample rate of $992\,\mathrm{Hz}$. The sample period is very close to $1\,\mathrm{ms}$ (1.0081), and when discussing model inputs and outputs we make the simplification of assuming 1 bin is $1\,\mathrm{ms}$. This means that while the $80 \times 1.0081\,\mathrm{ms}$ intervals actually sum to $80.65\,\mathrm{ms}$, we refer to the combined 80 bin interval as being $80\,\mathrm{ms}$ for simplicity.

An input to a model is a $5 \times 1024$-shaped array representing just over 1 second of stimulus and spike history—4 rows record the 4-LED stimulus, and the last row encodes the spike train as a binary sequence. For any given input, the ground-truth output is 1 of 81 possible events corresponding to a spike occurring in one of 81 intervals: 80 approximately $1\,\mathrm{ms}$ length intervals and the final infinite interval. The recording is broken into intervals along the time axis in order to form training, validation and test segments with a ratio 7:2:1. From a training, validation or test interval, a sample is formed from any sub-interval long enough to cover both input and output lengths ($1024+80$ bins).

## D.2  MODEL ARCHITECTURE

The overall model architecture can be summarized as: `head(gpt(cnn(input) + embed(cell_id)))`, where the head is either the `logmix` or `cat` head. The transformer architecture was described in Appendix C.2, and the CNN architecture is described below. The embedding is a standard 1-hot vector encoding of the integer cell ID $\in [0, 1611)$.

## D.3  CNN ENCODER

The CNN encoder shared by all spike prediction models takes a $5 \times 1024$-shaped input and produces a $64 \times 64$ shaped intermediate representation which is added to the cell ID embedding to form the input to the `gpt-a` or `gpt-b` transformers. The CNN architecture is described in Table 8. ResNet blocks (He et al., 2016) with an expansion factor of 4 form the core of the encoder. Squeeze and excite layers (Hu et al., 2018) were applied to the residual connections between ResNet blocks.

## D.4  OUTPUT HEADS

For the spike prediction task, the two output heads were implemented as linear functions (fully-connected layers) taking the last vector of the transformer output as input. The `logmix` outputs $64 \times 3 = 192$ values used to parameterize a mixture of 64 log-normal distributions. The `cat` head outputs 81 values, used to parameterize the categorical distribution over 81 intervals.

## D.5  TRAINING

A batch size of 1024 was used for all models. Models were trained over 2 epochs, which was empirically determined to be long enough to see all models exhibit overfitting. While 2 epochs may seem low, the $10\frac{1}{2}$ minutes of the training set produces over 1 billion unique timesteps ($10\frac{1}{2}$ minutes with 992 samples per second for 1611 cells), and each of these timesteps appear in multiple inputs per epoch. Cross-entropy loss is used for both output heads. This is a probability density calculation for the `logmix` head, and a probability mass calculation for the `cat` head. The probability mass calculation for the `cat` head is equivalent to a density scaled by the interval widths. While training, evaluations were carried out at regular training step intervals—every 5000 training steps. Other training settings shared by other experiments are described in Appendix E.

Table 8: Convolution layers of the stimulus and spike history encoder used for spike prediction, containing ~603 k trainable parameters.

| Layer | Input size | Kernels (length, channels, stride) | Output size |
|---|---|---|---|
| Conv 1 | 5×1024 | $[21, \quad 16, \quad 2] \times 1$ | $16 \times 512$ |
| Conv 2 | 16×512 | $[21, \quad 16, \quad 1] \times 1$ | $16 \times 512$ |
| ResNet blocks (downsampling) | $16 \times 512$ | $\begin{bmatrix} 1 & 128 & 1 \\ 7 & 128 & 2 \\ 1 & 64 & 1 \end{bmatrix} \times 3$ | $64 \times 64$ |
| ResNet block | $64 \times 64$ | $\begin{bmatrix} 1, & 128, & 1 \\ 7, & 128, & 1 \\ 1, & 64, & 1 \end{bmatrix} \times 1$ | $64 \times 64$ |

## D.6 EVALUATION

Four metrics are used to describe the performance of models at the spike prediction task: negative log-likelihood, Van Rossum distance, Schreiber similarity and smoothed Pearson correlation. The metrics are calculated on a held-out test interval of the RGC recording.

## D.7 NEGATIVE LOG-LIKELIHOOD

Negative log-likelihood (NLL) is the mean negative log-likelihood of the ground truth spikes given a model's output, calculated for every input-output snippet in the test set. This is a non-autoregressive metric. It is calculated in terms of probability mass, which involves integrating over the interval for the `logmix` head.

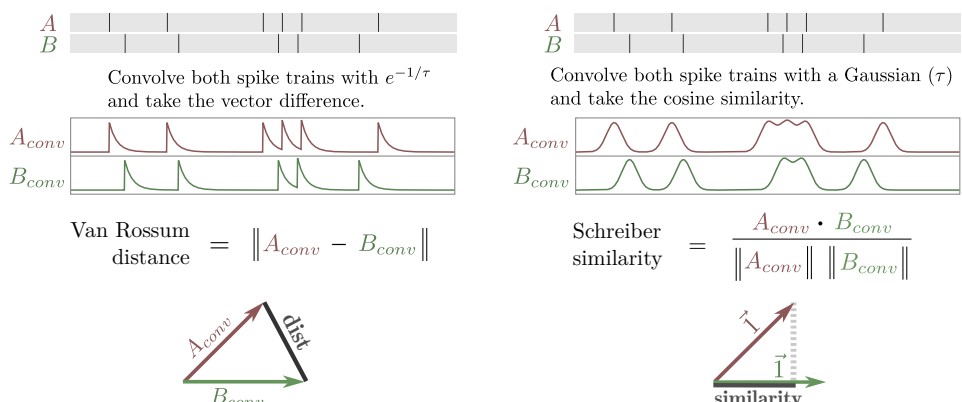

Figure 30: Calculation of Van Rossum distance (**left**) and Schreiber similarity (**right**) from two input spike trains. Both metrics employ distinct smoothing kernels to produce an intermediate vector. Van Rossum distance can be thought of as measuring the Euclidean distance between the vectors, while the Schreiber similarity assesses the angle between them. Figure taken from Doran et al. (2024).

### D.7.1 SPIKE TRAIN SIMILARITY/DIFFERENCE METRICS

The remaining 3 metrics are calculated after first producing a predicted spike train. A spike train is produced by evaluating a model autoregressively, shifting forward until the time of the next predicted

spike or the maximum stride of 80 bins. A model is given the first input snippet including the ground truth spikes. The median of the model's output distribution is used as the prediction for the next spike. For example, if there is a spike predicted in the 10th bin, then nine 0s and one 1 will be appended to the existing spike train. This process repeats until the stimulus is fully consumed.

Once a spike train is generated, Van Rossum distance, Schreiber similarity and smoothed Pearson correlation can be calculated. The process of calculating Van Rossum distance and Schreiber similarity is shown in Figure 30. Smoothed Pearson correlation is calculated as the well-known Pearson correlation after first smoothing with a Gaussian kernel. For all metrics, the smoothing parameter is set to $60\,\mathrm{ms}$, which is a length shown to be appropriate by Doran et al. (2024).

# E   TRAINING AND EVALUATION DETAILS

This appendix describes hyperparameters and other settings associated with training and evaluation. Appendix E.1 covers settings shared by all experiments. Settings specific to the spike prediction task were covered in Appendix D.5. Appendix E.2 covers settings associated with the remaining experiments.

## E.1   SHARED SETTINGS AND HYPERPARAMETERS

The following settings were shared across all training runs across all experiments.

**Software**. Pytorch 2.6.0 with Cuda 12.4.1 was used for training and evaluation.

**Optimizer**. All training runs use the AdamW optimizer described by (Loshchilov & Hutter, 2019). AdamW parameters ($\beta_1$, $\beta_2$, eps, weight decay) were set to (0.9, 0.99, $1 \times 10^{-5}$, 0.02)—chosen roughly in line with heuristics described by Howard et al. (2020). Weight decay was applied to all parameters except bias and embedding parameters.

**Precision**. Pytorch's automatic mixed precision and gradient scaling were used during training. In cases where greater numerical precision is needed, such as when calculating probabilities from hazard outputs, 32-bit floating-point precision was used.

**Loss function**. Most models were trained to minimize negative log-likelihood. The spike distance model that appears in Appendix D was trained to minimize the mean-squared error between the output and target representations. `thp-0` and `thp-1` used a loss term that is the sum of a negative log-likelihood term and a mean-squared error term (Zuo et al., 2020).

**Model choice**. The final models are selected from the checkpoints taken during training with the lowest validation loss.

**Learning rate scheduler**. The 1-cycle learning rate policy with 3 phases described by Smith & Topin (2017) was used.

**Maximum learning rate**. The maximum learning rate, which parameterizes the learning rate scheduler, was chosen by carrying out learning rate range tests as described by Smith (2017). The choice of learning rate is described in more detail below, in Appendix E.3.

**Early stopping**. An early-stopping strategy was used to reduce wasted compute. After at least 1/2 of the total training steps have been completed (when the learning rate is decreasing), early stopping is triggered if the smoothed validation loss (smoothed at 0.8) has not improved over 12 evaluations. Post-training, training and validation loss curves were inspected to ensure that early stopping was not triggered prematurely.

## E.2   SETTINGS FOR EXPERIMENTS FROM SECTIONS 3 TO 7

The approach to hyperparameter selection described in this section is used for the experiments from Section 3 onwards. The choice of settings such as batch size, training duration and learning rate is more involved for these experiments, as the heterogeneous combinations of models and datasets require a more sophisticated choice of hyperparameters.

**Training steps**. In order to compare training runs across datasets of different sizes, the number of epochs is not fixed. We set two upper limits: the number of samples drawn during training is capped at $2^{27}$ (~134 million samples), and the number of epochs is capped at 512. These limits are large enough so that both the larger and smaller datasets are trained for sufficient samples in order for training to converge (or overfit). The longest training set has $2^{25}$ events, and so each sample can be seen 4 times during training. For training set sizes smaller than $2^{19}$, samples will be seen 512 times during training. The combination of these caps with the batch size settings described next results in the training settings shown in Table 9.

**Batch size**. The wide range of training set sizes means that there isn't a single batch size suitable for all training runs. When the training set size is relatively large, a batch size of 1024 is used. This value was chosen as a batch size any smaller led to the overall training time becoming prohibitively long. As the size of the training set is reduced, this batch size eventually becomes unsuitably large

in comparison to the dataset size; for example, a whole epoch of the smallest training set would fit into a single batch, and we would no longer be performing stochastic gradient descent. To address this, batch size is reduced for the smaller datasets by limiting it to a maximum of $\frac{1}{128}$ of the training set size.

A limitation of having batch size vary with training set length is that it becomes another potential explanation for differences in performance. We argue that this does not detract from the results as the batch sizes that are paired with each training set size are representative of what a practitioner would use; for example, it would not be representative of common practice to train a model on the $2^{25}$ length training sets with a batch size of 4.

**Steps per evaluation**. Models are evaluated every 1024 training steps, or every epoch, whichever is sooner. Evaluating every epoch is too infrequent for the larger training sets. The effect of these settings is still to evaluate more frequently for smaller datasets; this is suitable as models quickly overfit on these datasets, and it is desirable to have more frequent evaluations in order to select the best-performing model parameters.

Table 9: Relationship between training set size and a number of settings: batch size, number of epochs, number of steps and number of evaluations. Batch size is capped at $\frac{1}{128}$ of the training set size. The number of epochs is capped at 512. $2^{27}$ samples are drawn, or until the epoch limit of 512 is reached. An evaluation is run every epoch or every 1024 steps, whichever is sooner. A consequence of these settings is that all configurations share the same number of steps.

| Train length | Batch size | Epochs | Steps | Evals |
|---|---|---|---|---|
| 1024 | 8 | 512 | 65536 | 512 |
| 2048 | 16 | 512 | 65536 | 512 |
| 4096 | 32 | 512 | 65536 | 512 |
| 8192 | 64 | 512 | 65536 | 512 |
| 16384 | 128 | 512 | 65536 | 512 |
| 32768 | 256 | 512 | 65536 | 512 |
| 65536 | 512 | 512 | 65536 | 512 |
| 131072 | 1024 | 512 | 65536 | 512 |
| 262144 | 2048 | 512 | 65536 | 512 |
| 524288 | 2048 | 256 | 65536 | 256 |
| 1048576 | 2048 | 128 | 65536 | 128 |
| 2097152 | 2048 | 64 | 65536 | 64 |
| 4194304 | 2048 | 32 | 65536 | 64 |
| 8388608 | 2048 | 16 | 65536 | 64 |
| 16777216 | 2048 | 8 | 65536 | 64 |
| 33554432 | 2048 | 4 | 65536 | 64 |

**Learning rate**. With multiple datasets, models and batch sizes, fixing a single learning rate would risk skewing results in favour of configurations that best suit the chosen learning rate. Instead of fixing a learning rate, we fix a strategy for selecting a learning rate. This is described in the next section.

### E.3   LEARNING RATE SELECTION

For all configurations of models and datasets, an individual maximum learning rate is chosen by using the learning rate range test introduced by Smith (2017). An issue with this approach is its sensitivity to noise in the loss curve generated by a sweep over learning rates; this noise makes it difficult to identify the region of steepest descent. We modify the approach slightly in order to increase robustness: for each configuration we carry out 8 learning rate sweeps. Averaging 8 sweeps is insufficient to smooth out the noise in the loss curves. Instead, we use Kalman filtering to estimate the expected change in loss at each learning rate and integrate this curve to obtain the final smoothed loss curve. This process generates smooth curves from which critical points can be more easily identified. From the smoothed loss curve, we choose a learning rate that is the geometric mean between the point of steepest descent and the point where the loss curve ends or begins to

increase. The selection of the two points—the point of steepest descent and the endpoint—is not always reliable, and so all selected learning rates were manually inspected, with a small number being manually overridden.

The results of this learning rate sweep for `gpt-a-cat` for the synthetic datasets are shown in Figure 31. The same results but for `rnn-nn` are shown in Figure 32. Interestingly, these results show that the learning rate identified by the range test varies very little between batch sizes on these datasets. To reduce the number of manual overrides, we take the median learning rate across batch sizes as the final learning rate and manually override this for configurations where it is too high or low.

Similar to the situation with batch size, having a learning rate vary across models and datasets introduces an extra factor that may account for differences in performance across configurations. This may be considered a limitation; however, we argue that to fairly compare models, training should be *best-effort*—that is, each model should be trained with the learning rate most appropriate for it and the dataset it is being trained on. In this sense, while the learning rate varies across models and datasets, the strategy for choosing the learning rate is held constant.

### E.4 COMPUTE RESOURCES

There were three computationally intensive steps for each experiment: learning rate sweeps, training and evaluation. Learning rate sweeps and model evaluation were carried out on a single workstation, with GPU, CPU and RAM specifications: Nvidia RTX 4090 GPU, AMD Ryzen Threadripper PRO 5975WX and 256 GiB RAM. Depending on the experiment, training was carried out on either this workstation or by using 10 Nvidia A40 GPUs from an internal cluster. The wall-clock durations for each of these three steps are reported below, separated by the sections where the results are used. The durations are affected by concurrent jobs that may have been sharing resources, and may include a tapering period where there is not enough remaining work to utilize all GPUs.

For the real-world datasets in Section 3 including the corresponding supplementary results, time taken for learning rate sweeps, training and evaluation was ~14 hours, ~129 hours and ~14 hours respectively. The learning rate sweeps and the evaluation were carried out on the workstation, and training was carried out on the cluster. For Sections 4 to 5, the same triplet of tasks took ~20 hours, ~223 hours and ~11 hours. For the modulo datasets (Section 7), it was ~16 hours, ~167 hours and ~7 hours. For the spike prediction experiment (Section 6), all three steps were run on the workstation. Time taken for the learning rate sweeps, training and evaluation was ~9 hours, ~122 hours and ~9 hours respectively.

Across all experiments, learning rate sweeps took ~59 hours on the workstation, training took ~122 hours on the workstation and ~519 hours on the cluster, and evaluation took ~41 hours on the workstation. We estimate that exploratory, failed or unused experiments used approximately 4 times more compute, combined.

### USE OF LARGE LANGUAGE MODEDLS

Tools utilizing large language models were used for spelling and grammar checking.

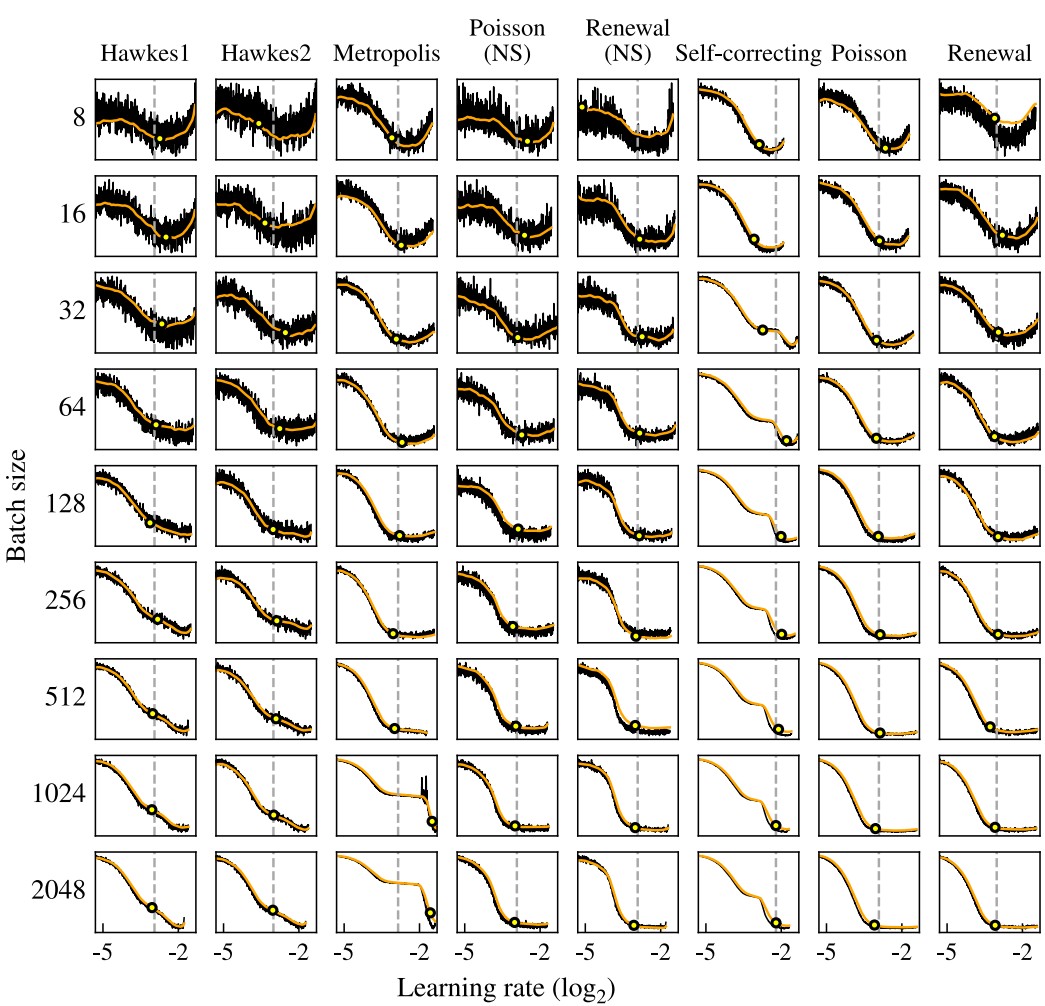

Figure 31: Learning rate sweep for `gpt-a-cat` for the synthetic datasets. The black trace (—) is the mean loss over 8 runs. The orange trace (—) is the Kalman smoothed loss. The yellow points (o) mark the geometric mean between the point of steepest descent and the point where the curve increases or ends—the proposed learning rate. The dashed vertical grey line marks the median proposed learning rate across batch sizes. As the proposed learning rate varies little between batch sizes, in this work, we choose the median learning rate across batch sizes to be used for all batch sizes. We manually select the learning rate for a configuration if the median learning rate is unsuitably high or low.

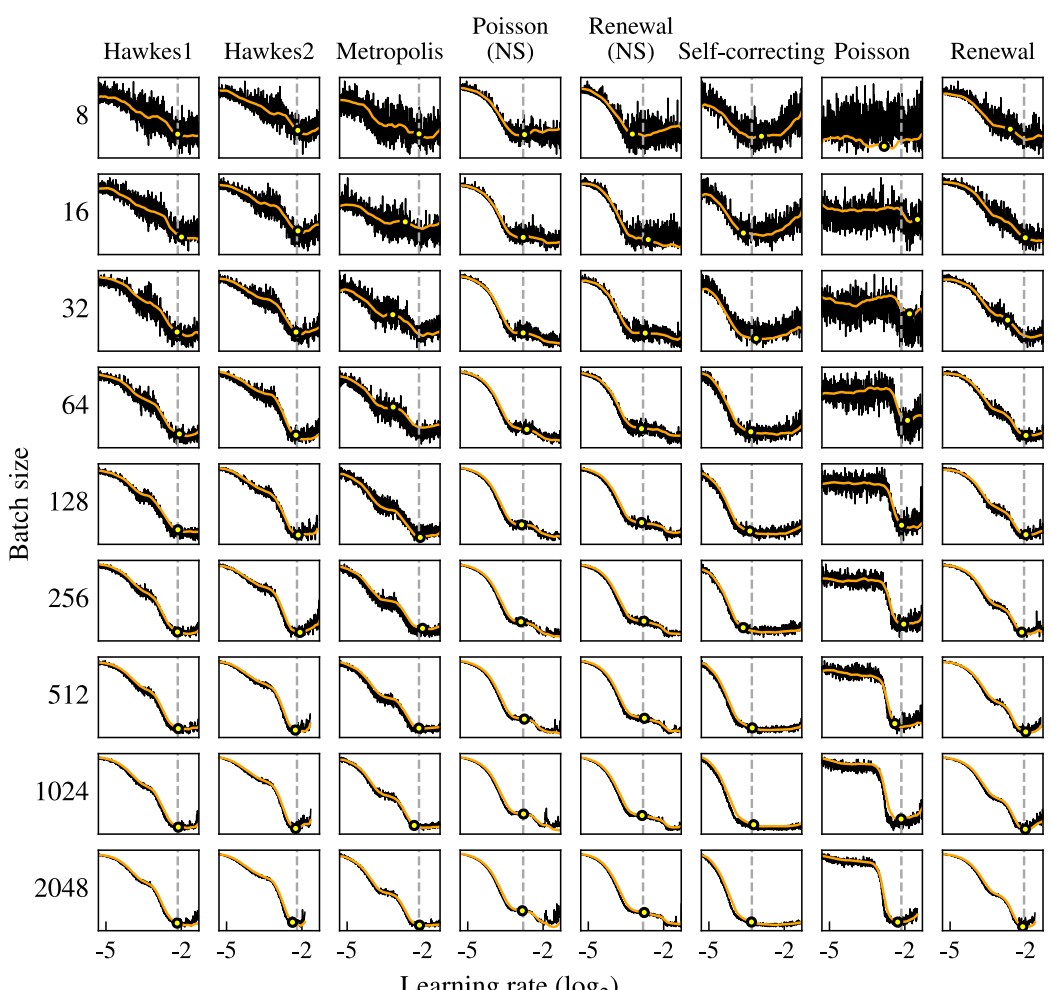

Figure 32: Learning rate sweep for `rnn-nn` for the synthetic datasets. The black trace (—) is the mean loss over 8 runs. The orange trace (—) is the Kalman smoothed loss. The yellow points (o) mark the geometric mean between the point of steepest descent and the point where the curve increases or ends—the proposed learning rate. The dashed vertical grey line marks the median proposed learning rate across batch sizes.

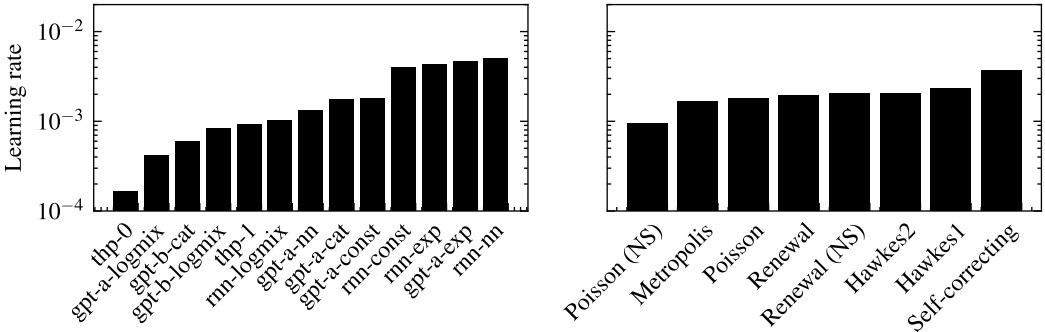

Figure 33: **Left:** mean learning rate used for each model (mean over synthetic datasets). **Right:** mean learning rate used for each of the synthetic datasets (mean over all models).

