# OpenReview forum: "Categorical Distributions are Effective Neural Network Outputs for Event Prediction"
_ICLR.cc/2026/Conference — ICLR 2026 Conference Withdrawn Submission_

### Official Review · Reviewer_dsSi · 2025-10-22

**Soundness:** 2
**Presentation:** 1
**Contribution:** 2
**Rating:** 2
**Confidence:** 4

**Summary:**

disclaimer: i dont use any LLMs to write this review. Everything is purely from my understanding from reading this paper (without any help of LLM)

The paper challeges the traditional way of predict next event given event history , which typically treat the next event as a continuous random variable ( i.e. E_{t~f(t)} [t] ) where f(t) is the probability density of next event. Instead the author(s) proposed to use discrete N bins for modeling the distribution of next event. They found such proposed method works well on quite a lot of synthetic and real benchmarks esp. training set size is large. They also find a real application for using discretized bin for spike prediction – the data from chicken RGCs.

**Strengths:**

The main strength is it provides a different angle to look at next event prediction.
The authors have conducted many experiments to support their claim as suggested by the title CATEGORICAL DISTRIBUTIONS ARE EFFECTIVE NEURAL NETWORK OUTPUTS FOR EVENT PREDICTION.

**Weaknesses:**

originality : it is okay, but it can be improved with some theoretical insight of the gain of NLL (or LL) with the categorical distribution.

Quality: The authors should include other evaluation metric such as MAE, MSE than NLL in the figure 3. For Table 1: I am not sure if the proposed method is better.  I also think readers will appreciate more if the authors elucidate with some theoretical underpinnings on the threshold of the LL increases as training set size increases in Figure 3.

Clarity: I am not sure if it is the best way to write this paper. I thought there is no proposed method contained and it was purely an assignment from class project. In the end I figured out the flow of this paper.

Significance. It can be interesting to TPP community. However, it is rather strange to discretize it since TPP and its continuous characterization of next event are so elegant and well established.  As the authors mention that the current model only support TPP without marks, limiting its significance.

**Questions:**

1.	How do you select N ? I know you mention folling Kvamme & Borgan (2021).

2.	Another comment is one application Chicken RGCs is fairly limiting. The authors can incorporate 1-2 more.

---

> ### Author Response · Authors · 2025-11-19
>
> We sincerely thank the reviewer for providing feedback and raising valuable concerns.
>
> > **..it can be improved with some theoretical insight of the gain of NLL (or LL) with the categorical distribution.**
>
> I believe the text may already answer this question. The other output heads tested either have a restricted distribution shape (parameterized by only 1 or 2 parameters) or have unstable training when fitting narrow peaks. Both limitations are discussed in Appendix C (L1700-1725). The instability of the lognormal mixture is analogous to the well-known issues of singularities encountered when fitting mixtures of Gaussians with the expectation-maximization algorithm. The categorical distribution has a flexible shape and does not encounter training instability.
>
> > **The authors should include other evaluation metric such as MAE, MSE than NLL in the figure 3.**
>
> MAE results are in Appendix A (Figure 13 and 15).
>
> > **For Table 1: I am not sure if the proposed method is better.**
>
> The proposed method scores well, obtaining the best (lower) NLL on 7 of 12 datasets, with statistically significant differences (see Table 4 for CI intervals). *We propose to* include CI intervals for the differences reported in Table 1.
>
> Despite the proposed method scoring well, a claim of the paper is that these datasets bias evaluations towards models that are well specified to small datasets and are not representative of applications where larger amounts of data are available.
>
> > **it is rather strange to discretize it since TPP and its continuous characterization of next event are so elegant and well established.**
>
> For evaluating with the continuous datasets, the categorical output is interpreted as a piecewise-continuous distribution, not a discrete distribution. Does this clarification help?
>
> I am curious if you find the spike prediction task strange. Most practical applications involve discrete time measurements, and we believe that the field's lack of consideration of discrete targets is a significant gap in the literature.
>
> > **How do you select N ? I know you mention following Kvamme & Borgan (2021).**
>
> We found N=128 to be effective across all datasets studied.
> It was chosen as the power of 2 less than the number of parameters in the mixture of lognormals head (so that the categorical model would not have greater capacity in terms of model parameters). We did not perform a systematic hyperparameter search over N. *We propose to* add to the appendix an experiment varying the number of bins to demonstrate the effect of this hyperparameter.
>
> > **As the authors mention that the current model only support TPP without marks, limiting its significance.**
>
> The output distribution is fully compatible with predicting marks, it is simply not explored in this work. Marks are typically modelled with a second categorical output.
>
> > **Another comment is one application Chicken RGCs is fairly limiting. The authors can incorporate 1-2 more.**
>
> Most prior neural-TPP works treat the "real-world" sequences (tweets, taxi pickup times, etc) as applications, of which 14 are covered in this work. The spike prediction task is an important contribution due to the lack of true application-grounded tasks used in the field.

---

### Official Review · Reviewer_zYmH · 2025-10-27

**Soundness:** 2
**Presentation:** 1
**Contribution:** 2
**Rating:** 2
**Confidence:** 3

**Summary:**

The paper proposes modeling inter-event times in temporal point processes by discretizing the time distribution into quantile-based bins and predicting a categorical distribution over these intervals. The authors argue that this piecewise-constant representation is better aligned with real-world datasets that exhibit discrete temporal spikes. The authors additionally examine how training set size impacts model performance and introduce several new datasets.

**Strengths:**

- Focus on a practically motivated issue: real event-time data often contains discrete patterns (e.g., NYC taxi), where density-based likelihoods may be unstable.
- Clear and intuitive modeling approach.
- Contribution of new datasets may facilitate future research.

**Weaknesses:**

- Theoretical formulation lacks rigor and clarity.
  - The hybrid model combining discrete time bins with continuous likelihoods is not formally defined, leaving ambiguity in the underlying probability space.
  - The suitability of the negative log-likelihood objective is questionable in this mixed setting. For example, in the NYT dataset, likelihood can be artificially inflated by collapsing predictive variance toward zero at discrete timestamps, making likelihood comparisons unreliable.
- Table 1 reports only NLL, omitting standard point-forecasting metrics (e.g., MAE, RMSE) widely used in time-series forecasting to avoid density anomalies.
- The paper does not compare against simple continuous regression losses (MSE/MAE), which is necessary to establish whether the proposed discretization offers real benefits over standard formulations.
- The paper does not compare against recent NN-based models [2, 3, 4].
- Claims regarding improved loss behavior, sensitivity to dataset size, and the impact of newly collected datasets are mixed, resulting in limited analysis of each claim and making it difficult to assess their individual contributions.
- The contribution may be overstated relative to prior work employing quantile-based timestamp transformations in TPPs [1].
- Writing can be strengthened by correcting grammatical errors, eliminating repetition, and avoiding unsupported generalizations (e.g. "This type of distribution is not uncommon", "For this dataset, not only are events from a real-world process, the task itself replicates the requirements of a real-world task.").

[1] Learning Quantile Functions for Temporal Point Processes with Recurrent Neural Splines (2022)

[2] Recurrent marked temporal point processes: Embedding event history to vector (2016)

[3] The neural hawkes process: A neurally self-modulating multivariate point process (2017)

[4] Latent ordinary differential equations for irregularly-sampled time series (2019)

**Questions:**

See weaknesses.

---

> ### Author Response · Authors · 2025-11-15
>
> Thank you for the review. Below I address each of the points you raised under Weaknesses.
>
> > **The hybrid model combining discrete time bins with continuous likelihoods is not formally defined, leaving ambiguity in the underlying probability space.**
>
> For the continuous-time case, the output distribution is formally defined (Section 2). Can you clarify what you mean by the hybrid discrete-continuous combination and where the ambiguity lies?
>
> > **The suitability of the negative log-likelihood objective is questionable in this mixed setting. For example, in the NYT dataset, likelihood can be artificially inflated by collapsing predictive variance toward zero at discrete timestamps, making likelihood comparisons unreliable.**
>
> I agree that likelihood as a metric is problematic. For this reason, the paper never solely reports likelihood (see Appendix A for paired MAE results). For realistic tasks, both MAE and likelihood can be unsuitable; Section 6 is dedicated to introducing a novel event prediction task with application-grounded metrics.
>
> Which aspect of the evaluation do you consider mixed? The evaluation setting treats all models identically and does not prevent any model from outputting narrow and high densities (beyond their own structures). The mixture of lognormals head can indeed output such densities, although it leads to training instability.
>
> > **Table 1 reports only NLL, omitting standard point-forecasting metrics (e.g., MAE, RMSE) widely used in time-series forecasting to avoid density anomalies.**
>
> MAE results are in Table 6.
>
> > **The paper does not compare against simple continuous regression losses (MSE/MAE), which is necessary to establish whether the proposed discretization offers real benefits over standard formulations.**
>
> MAE results are in Table 6 and in Figure 11, Figure 13 and Figure 15.
>
> > **The paper does not compare against recent NN-based models [2, 3, 4].**
>
> We test the output heads used by all three methods you refer to. Du et al. (2016) use an RNN with exponential hazard and Mei et al. (2017) use an RNN with softplus hazard. Rubanova et al. (2019) does not present a next-event prediction model, but for the time series predictions, their model does utilize an inhomogeneous Poisson output, which is covered in our work (the constant hazard output).
>
>
> > **Claims regarding improved loss behavior, sensitivity to dataset size, and the impact of newly collected datasets are mixed, resulting in limited analysis of each claim and making it difficult to assess their individual contributions.**
>
> Can you specify which results appear mixed and how those results imply limited analysis of each claim?
>
>
> > **The contribution may be overstated relative to prior work employing quantile-based timestamp transformations in TPPs [1].**
>
> The paper you refer to proposes a fundamentally different approach: a continuous parametrization of a density (implicitly, via its quantile function). Additionally, it does not address issues with datasets used for evaluating TPPs, which is a main contribution of our work.
>
>
> > **Writing can be strengthened by correcting grammatical errors, eliminating repetition, and avoiding unsupported generalizations (e.g. "This type of distribution is not uncommon", "For this dataset, not only are events from a real-world process, the task itself replicates the requirements of a real-world task.").**
>
> I am eager to improve writing quality; however, for the examples you point to, I believe they are appropriately supported, grammatically correct, and without repetition.

---

> ### Comment · Reviewer_zYmH · 2025-11-28
>
> Thank you for the detailed feedback.
>
> **Hybrid model**
>
> The paper analyzes datasets composed of a mixture of distributions, one continuous and one discrete (spike-based). During likelihood computation, the model evaluates $P(Y \mid X)$, which corresponds to a probability density function in the continuous case and to a probability mass in the discrete case. This distinction and the resulting dual interpretation are not clearly explained in the current manuscript.
>
> **MAE**
>
> Table 6 does not show a consistent improvement in MAE compared to the baselines. The proposed method outperforms alternative approaches in only a minority of the reported cases.
>
> **MAE regression losses**
> The relevant baseline is an RNN or Transformer model trained using the MAE loss for timestamp prediction. This approach is specifically tailored to minimize MAE. The corresponding likelihood can be derived by interpreting the MAE objective as the negative log-likelihood of a standard Laplace distribution.

---

> > ### Author Response · Authors · 2025-12-01
> >
> > > **The paper analyzes datasets composed of a mixture of distributions, one continuous and one discrete (spike-based). During likelihood computation, the model evaluates $P(Y \mid X)$, which corresponds to a probability density function in the continuous case and to a probability mass in the discrete case. This distinction and the resulting dual interpretation are not clearly explained in the current manuscript.**
> >
> > For no dataset is there a dual interpretation of the likelihood. For the spike prediction task, all models are evaluated with probability mass, and this is explained on lines 333-334.
> >
> > > **Table 6 does not show a consistent improvement in MAE compared to the baselines. The proposed method outperforms alternative approaches in only a minority of the reported cases.**
> >
> > The model that scores highest over the most datasets _is_ rnn-cat, the RNN with categorical head. Can you elaborate as to how this result makes the proposed method's scores insufficient?
> >
> > > **MAE regression losses The relevant baseline is an RNN or Transformer model trained using the MAE loss for timestamp prediction. This approach is specifically tailored to minimize MAE. The corresponding likelihood can be derived by interpreting the MAE objective as the negative log-likelihood of a standard Laplace distribution.**
> >
> > MAE as a metric for models trained to minimize NLL is a common metric for evaluating event prediction models. More generally, it is common to evaluate models on metrics that are not identical to the loss signals used to train the models.

---

### Official Review · Reviewer_1ngV · 2025-11-01

**Soundness:** 2
**Presentation:** 3
**Contribution:** 3
**Rating:** 4
**Confidence:** 4

**Summary:**

The authors present three contributions to the problem of predicting event distributions. First they show that a categorical distribution (rather than a continuous specification) is effective. Second, they show that differences between models (with continuous and discrete distributions) get smaller as the size of the dataset increases. Third, they introduce three datasets, two synthetically generated so that in one performance is less dependent on sample size while in the other sample size is correlated with expected performance. The third dataset is for a spike prediction task motivated by retinal prosthetics.

**Strengths:**

This is mainly an experimental paper that seeks to demonstrate scenarios in which a discrete distribution is useful to characterize event distributions and the circumstances under which it is expected to perform better than a continuous specification. For the most part, the experiments are well motivated and justified, and the experiments are convincing and sufficiently detailed in the supplementary material.

**Weaknesses:**

The main weakness of the paper is that it seems to show two fairly evident phenomena event prediction, namely that in general, given sufficient sample size continuous, discrete and implicit (via sampling) event distribution estimates perform similarly and well specified models (meaning those that match the underlying process) perform better. This naturally does not take away the effort put by the authors in demonstrating it empirically.

Although the introduced artificial datasets help illustrate the points above, it is less clear whether they further our understanding of models for event distribution prediction, especially, in practical settings where sample sizes and knowledge of the underlying process are usually limited.

Understanding that this is an experimental contribution, it is worth mentioning that the technical contributions are minimal.

**Questions:**

- Results in Table 1 are provided with error bars in Table 4. Consider indicating the significance of the differences in Table 1.
- The motivation for the Metropolis lognormal dataset is understood, however, it is unclear whether it is a representation of something that is likely to happen in practice. After all, none of the other real-world datasets being considered seem to support it. In a sense, the dataset seems almost to contrived.
- Consider having results for the spike prediction task with a range of sample sizes.

---

> ### Author Response · Authors · 2025-11-19
>
> We sincerely thank the reviewer for providing feedback and raising valuable concerns.
>
> > **The main weakness of the paper is that it seems to show two fairly evident phenomena event prediction, namely that in general, given sufficient sample size continuous, discrete and implicit (via sampling) event distribution estimates perform similarly and well specified models (meaning those that match the underlying process) perform better.**
>
> We would like to suggest a different framing that may partially ameliorate this weakness of contribution:
>
> The basic phenomena were *utilized* to show how commonly used datasets in the field of neural-TPPs bias evaluations towards models that are well specified to small datasets. It is important for the field to consider the consequences of both larger datasets and datasets that are motivated by applications. One consequence is that the simple categorical output can be competitive.
>
>
> > **The motivation for the Metropolis lognormal dataset is understood, however, it is unclear whether it is a representation of something that is likely to happen in practice. After all, none of the other real-world datasets being considered seem to support it. In a sense, the dataset seems almost to contrived.**
>
> The "broken" Metropolis sampler is intended to replace the source of randomness for _any_ 1D distribution. The lognormal distribution was chosen as an example so that the lognormal mixture output was not disadvantaged.
>
> Connection to real-world processes: a real-world event sequence may have a certain distribution in aggregate, say Poisson, but the underlying dynamics leading to the aggregate behaviour may be decodable to a greater extent than the shape of the Poisson distribution. The broken Metropolis sampler can be used to test if models predict with performance beyond an aggregate distribution.
>
> We realize that this motivation is not clearly presented in the paper, and *we propose to* add the explanation to the relevant section (Section 5).
>
>
> > **Results in Table 1 are provided with error bars in Table 4. Consider indicating the significance of the differences in Table 1.**
>
> *We propose to* add confidence intervals for the differences in Table 1.
>
> > **Although the introduced artificial datasets help illustrate the points above, it is less clear whether they further our understanding of models for event distribution prediction, especially, in practical settings where sample sizes and knowledge of the underlying process are usually limited.**
>
> Regarding the modulo sequences. There already exists a battery of commonly used synthetic datasets for continuous-time predictions, but no such datasets for discrete-time predictions. If you agree that the spike prediction task demonstrated the significance of considering a discrete output space, then we argue that having a set of synthetic discrete-time datasets is useful.
>
> Secondly, the sequences have a form that allows analysis in terms of Ethayarajh et al.'s (2022) v-usable information, giving a theoretical basis for discussing dataset difficulty. Figure 27 demonstrates this link. By testing models across the family of modulo sequences, a practitioner can understand how well their models may perform over a range of dataset difficulties. *We propose* to make the point regarding v-usable information more explicit in the main text.

---

### Official Review · Reviewer_uxbv · 2025-11-01

**Soundness:** 2
**Presentation:** 1
**Contribution:** 2
**Rating:** 2
**Confidence:** 2

**Summary:**

The paper investigates the use of categorical distributions as neural network outputs for temporal point process modeling and event time prediction. The authors propose discretizing inter-event times into quantile-based intervals and predicting a categorical probability for each bin. They state that it can be an effective alternative to traditional continuous-density distributions (e.g., exponential or lognormal mixtures). Results indicate that categorical outputs perform competitively with, or better than, lognormal mixtures and other TPP models. They show that the proposed method benefits from the data with inherent discreteness or spikes. The paper concludes that output choice and dataset structure jointly determine performance trends in neural TPPs.

The authors propose an interesting methodology that replaces standard continuous output distributions with interval-based categorical representations. They demonstrated that this approach can be effective for event-time distributions with sharp peaks or for clear discrete structures. However, the research lacks the overall depth of analysis.
The paper’s main weakness is that it centers on comparing lognormal mixture models and their shortcomings, while the inclusion of other TPP models (NHP, RMTPP, THP) lacks meaningful qualitative analysis that directly compares their behavior to the categorical distribution method. The section on the Metropolis Lognormal Event Dataset is methodologically sound, but it does not clearly relate to the categorical distribution method. The paper should clarify how this section is connected to the evaluation of categorical outputs.
Similarly, the Event Sequences from Modulo Addition section lacks clarity in its connection to the stochastic foundations of TPP modeling, since the process is described as deterministic except for the initial point parameters a and v. Clarify how this example relates to the overall aim of evaluating categorical representations in stochastic TPP modeling, such as by specifying whether the deterministic design exposes strengths or limitations of the proposed methodology. Additionally, some terms, such as “usable information,” are introduced without formal definition or theoretical grounding, which weakens the analysis.
The paper lacks focus on the proposed discretization method. It should more directly address clarifying its theoretical boundaries, comparing it with other TPP family models, and avoiding lateral discussion of general neural TPP training behaviors.

The paper is poorly written and difficult to follow. The abstract does not clearly establish the main focus. It is unclear whether the core contribution lies in the categorical distribution formulation or in the introduction of new datasets, and how these two aspects are conceptually connected. The overall narrative lacks coherence, and transitions between sections are abrupt, making the paper hard to read and potentially limiting its impact and accessibility.
Several parts of the experiments appear disjointed and do not build on one another consistently, which reduces the clarity of the study's overall findings. The figures are challenging to interpret, especially Figure 11, where plots are arranged by methods rather than by dataset dimensionality, making it difficult to compare methods within the same dimension.
The discussion of results is also overly superficial. The authors mostly describe outcomes without analyzing the underlying causes or implications. Overall, while the experiments are extensive, the presentation quality is well below ICLR standards.

The authors present an interesting and intuitive idea of using categorical discretization for event time prediction and demonstrate two concrete cases where this approach is well justified (Spike prediction and natively discrete problems). However, the overall exploration of the method’s capabilities and its theoretical or empirical boundaries is limited.
The relationship between model performance and the number of categorical bins is not analyzed, leaving open whether the method is sensitive or robust to discretization granularity. Comparisons with alternative TPP model families (e.g., NHP, RMTPP, diffusion- or ODE-based approaches) are only superficial, lacking a deeper qualitative discussion of why categorical heads might outperform or underperform across different regimes.
The paper fails to define which dataset characteristics justify the method's use. It is unclear when discretization is beneficial (e.g., for quantized or multimodal peak datasets) or harmful (e.g., for smooth continuous distributions). Additionally, several experiments do not directly address the main hypothesis. Section 5 (Metropolis Lognormal) prioritizes scalability rather than the categorical formulation, while Section 7 (Modulo Addition), though conceptually related, is poorly integrated and weakly motivated.
The paper fails to analyze how the number of bins affects training efficiency or model complexity, and provides no actionable guidance for practitioners. As a result, despite the originality and potential impact of the central idea, the work is underdeveloped and fragmented.

**Strengths:**

- Conceptual novelty: The paper introduces an interesting idea to represent inter-event times via categorical distributions instead of traditional continuous outputs. It provides a fresh and intuitive perspective on temporal point process modeling by focusing on probability mass rather than density.
- Comprehensive empirical evaluation: The authors conduct extensive experiments across a wide range of real-world and synthetic datasets. The results consistently demonstrate that categorical outputs are competitive with, and sometimes superior to, lognormal mixtures. It has been explicitly shown for datasets with discrete or multimodal event-time structures. The analysis of how training set size affects model performance is particularly insightful.
- New benchmark datasets: Three newly introduced datasets enrich the TPP research landscape, which can be useful for future TPP research:
Metropolis Lognormal Dataset — a synthetic benchmark for studying data scaling effects and model size sensitivity.
Retinal Spike Prediction Dataset — a biologically grounded benchmark linking TPPs with neuroscience applications.
Modulo Addition Family — synthetic discrete-time datasets enabling controlled evaluation of model generalization.
- Practical robustness and interpretability: The categorical formulation avoids the numerical instabilities of mixture models. Also, it naturally aligns with tasks with discretized or quantized observations (e.g., logs, sensor data, neural spikes).
- Useful insights for the TPP community: The paper highlights that differences between neural TPP models are often connected to differences in training set size and regularization, rather than architectural complexity alone. It also raises the important discussion of evaluating probability mass within intervals rather than the continuous function. This perspective could influence future research on loss design for temporal models.

**Weaknesses:**

- Lack of focus and narrative coherence: The paper attempts to address multiple directions — categorical outputs, dataset scaling, and new synthetic processes. However, these threads are not well integrated. As a result, the core contribution (the categorical output formulation) is diluted by other sections, such as Metropolis Lognormal and Modulo Addition, which contribute little to the central research question.
- Insufficient theoretical depth: The paper’s arguments about regularization effects and “usable information” remain qualitative and speculative. No formal analysis (e.g., entropy measures, mutual information, or bias–variance trade-offs) is provided to support the claimed advantages of discretization.
- Limited comparative analysis: Although several baselines are included, the main comparison is conducted with the lognormal mixture models; other models are covered superficially. The discussion does not explore why categorical heads succeed or fail under different conditions, except for a couple of concrete examples.
- Missing sensitivity and ablation studies: The method’s dependence on the number of bins, binning strategy, and dataset properties (e.g., presence of discrete peaks) is not analyzed. Without this, it is unclear how generalizable or robust the approach is across domains.
- Poor presentation quality: The writing is fragmented and difficult to follow. Figures, particularly Figure 11, are confusingly organized and under-labeled, making interpretation challenging. The discussion of results is mostly descriptive rather than analytical.
- Reproducibility issue: Although the paper explicitly states that it provides supplementary code and datasets for full reproducibility, I was unable to find the link in the submission. Given that reproducibility is one of the claimed strengths, the absence of an accessible codebase substantially undermines the credibility of the results.

**Questions:**

- Reproducibility: You mention that supplementary code and datasets are provided for all experiments, but I was unable to find a repository link or reference in the submission. Could you kindly clarify where the code and data will be available?
- Sensitivity to bin count: How does the model’s performance depend on the number of categorical bins? Is there a principled way to choose this number based on dataset characteristics?
- Generalization boundaries: For which types of datasets (e.g., continuous vs. discrete, unimodal vs. multimodal) do you expect the categorical output to perform best? Have you tested cases where discretization could harm performance?
- Relation to existing methods: Could you discuss how your approach compares empirically and conceptually with other modern neural TPPs, except for the lognormal mixture families, such as Neural Hawkes Processes, RMTPP, THP, Neural ODE-based, or diffusion-based TPPs?
- Regularization hypothesis: You argue that different output structures act as implicit regularizers. Your work would benefit from quantitative evidence (e.g., learning curves, entropy of predicted distributions, or generalization gaps) supporting this claim.
- Theoretical grounding: The paper frequently uses the term “usable information” without a formal definition. Could you possibly connect this concept to established measures (e.g., mutual information)?
- Section relevance: How do the Metropolis Lognormal and Modulo Addition datasets sections connect to validating the categorical approach? They seem to be only weakly related to the main hypothesis. Could you elaborate on how these experiments strengthen your central claim?
- Computational aspects: How does the categorical formulation affect training speed and memory compared to mixture-based models? Based on your experience, is the categorical head not only more stable but also more efficient?

---

> ### Author Response · Authors · 2025-11-17
> **Response summary**
>
> We sincerely thank the reviewer for providing comprehensive feedback and raising valuable concerns.
>
> We address the reviewer's points over two further comment blocks. We have grouped related concerns together for brevity.
>
> In the first comment, we address four points raised by the reviewer:
>
>   * ambiguity of the core contribution
>   * ambiguity of how the new datasets relate to the core contribution
>   * the lack of formal definition of "usable information"
>   * insufficient comparison with other TPP models
>
> In the second comment, we address the remaining points and questions:
>
>   * insufficient theoretical depth
>   * lack of quantitative evidence for the regularization hypothesis
>   * lack of analysis of method dependence on binning strategy and dataset properties
>   * questions about generalization, training speed and reproducibility
>
> In addressing the reviewer's concerns, we propose to:
>
>   * clarify the core contributions in the introduction
>   * rephrase "usable information" with "performance gain"
>   * add learning curves
>   * add an experiment that sweeps over the bin count
>   * move the section Distribution Flexibility (L1708-1722) to the main body

---

> > ### Author Response · Authors · 2025-11-17
> > **Response (part 1/2)**
> >
> > > **It is unclear whether the core contribution lies in the categorical distribution formulation or in the introduction of new datasets, and how these two aspects are conceptually connected.**
> > > **the core contribution (the categorical output formulation) is diluted by other sections, such as Metropolis Lognormal and Modulo Addition.**
> >
> > We aimed to present _two_ core contributions:
> >
> >   * highlight and address limitations of existing datasets
> >   * highlight the effectiveness of the categorical output
> >
> > *We propose to* add this clarification to the introduction.
> >
> > The connection between the two is that existing datasets are surprisingly skewed to favor models that perform well on smaller training sets.
> >
> > > **The section on the Metropolis Lognormal Event Dataset is methodologically sound, but it does not clearly relate to the categorical distribution method.**
> >
> > The "broken" Metropolis sampler can generate synthetic datasets that avoid an issue experienced with existing synthetic datasets—namely that small training sets are sufficient to reach near-optimal performance; in effect, powerful models can decode those sequences up to the noise from the pseudo-random number generator (see the "theoretical best" line in Figure 3 and Figure 12). The "broken" Metropolis sampler can be used to generate any 1D distribution. This work presents one example: a lognormal distribution. On this dataset, larger models and those with less constrained output heads (categorical and lognormal mixture) perform better.
> >
> > > **the Event Sequences from Modulo Addition section lacks clarity in its connection to the stochastic foundations of TPP modeling...**
> >
> > The modulo addition event sequences have two purposes.
> >
> > Firstly, they capture the form of the spike prediction task (which demonstrated how applications may naturally benefit from discrete output distributions). They are further beneficial as they have controllable difficulty and can be formally analyzed (see Appendix B.7).
> >
> > Secondly, the sequences have a form that allows analysis in terms of Ethayarajh et al.'s (2022) v-information, giving a theoretical basis for dataset difficulty. Figure 27 demonstrates this link.
> >
> > > **some terms, such as “usable information,” are introduced without formal definition or theoretical grounding, which weakens the analysis.**
> >
> > > **The paper frequently uses the term “usable information” without a formal definition. Could you possibly connect this concept to established measures (e.g., mutual information)?**
> >
> > *We propose to* change phrases such as "quickly plateau in terms of usable information" with "quickly plateau in terms of model performance gains". The original motivation for the "usable information" phrase was the eventual connection to v-usable information (Ethayarajh et al., 2022) (see Figure 27). We agree with the reviewer that in its current usage, it is not clear nor formally grounded.
> >
> > > **The paper’s main weakness is that it centers on comparing lognormal mixture models and their shortcomings, while the inclusion of other TPP models (NHP, RMTPP, THP) lacks meaningful qualitative analysis that directly compares their behavior to the categorical distribution method.**
> >
> > > **Comparisons with alternative TPP model families (e.g., NHP, RMTPP, diffusion- or ODE-based approaches) are only superficial, lacking a deeper qualitative discussion of why categorical heads might outperform or underperform across different regimes.**
> >
> > We agree that the lognormal mixture is given extra focus. This is due to it being more competitive (sometimes substantially) than the alternative approaches (see Table 5). However, all models are extensively compared. Table 1, which focuses on the lognormal mixture, has corresponding tables in Appendix A that include all models (Table 5 and 6). The section Distribution Flexibility (L1708) has a qualitative discussion of the heads. *We propose to* move this section to the main body.
> >
> > > **Could you discuss how your approach compares empirically and conceptually with other modern neural TPPs, except for the lognormal mixture families, such as Neural Hawkes Processes, RMTPP, THP, Neural ODE-based, or diffusion-based TPPs?**
> >
> > All experiments except for the spike prediction task include the RMTPP and THP models. They are labeled `rnn-exp` and `thp-0` and are linked to the original papers in Table 2. This work focuses on output heads, and the Neural Hawkes Process shares the same output head as THP (softplus). Neural ODEs are concerned with the evolution of hidden state, usually for predicting values _at_ future time points, whereas this work focuses on output distributions for autoregressive next event prediction.  Ludke et al.'s (2024) diffusion-based approach does not operate as autoregressive next event prediction, but rather outputs multiple events in output windows.

---

> > ### Author Response · Authors · 2025-11-17
> > **Response (part 2/2)**
> >
> > > **Insufficient theoretical depth: The paper’s arguments about regularization effects and “usable information” remain qualitative and speculative. No formal analysis (e.g., entropy measures, mutual information, or bias–variance trade-offs) is provided to support the claimed advantages of discretization.**
> >
> > > **Regularization hypothesis: You argue that different output structures act as implicit regularizers. Your work would benefit from quantitative evidence (e.g., learning curves, entropy of predicted distributions, or generalization gaps) supporting this claim.**
> >
> > We agree that the strength of the claim would be improved with additional evidence. *We propose to* add learning curves to the appendix for the datasets in Figure 3.
> >
> > We argue that you can observe bias tradeoffs in figures such as Figure 3. For example, for the categorical or lognormal mixture heads, swapping to larger base models (`rnn` to `gpt-a` to `gpt-b`) reduces performance for less training data but improves performance for larger training data.
> >
> > > **The method’s dependence on the number of bins, binning strategy, and dataset properties (e.g., presence of discrete peaks) is not analyzed. Without this, it is unclear how generalizable or robust the approach is across domains.**
> >
> > Addressing the three aspects separately:
> >
> >   * Number of bins. In terms of robustness, a single bin count (128) was effective across all experiments. Nevertheless, we agree that an experiment varying bin count would be useful (see the next reply).
> >   * Binning strategy. The bin edges are implied by the bin count and the training set quantiles. I might be misinterpreting—do you wish to see effects of different strategies for estimating quantiles?
> >   * Presence/absence of discrete peaks. The success of the categorical output is not dependent on the presence of discrete peaks; for example, it is competitive on the continuous synthetic datasets of sufficient training set size (Figure 3 and 12). The method is clearly effective across a wide range of datasets (Table 1, 5 and 6, Figure 3, 6, 11-16).
> >
> > > **Sensitivity to bin count: How does the model’s performance depend on the number of categorical bins? Is there a principled way to choose this number based on dataset characteristics?**
> >
> > We agree that the paper would benefit from an experiment exploring the effect of the number of bins. *We propose to* add to the appendix an experiment varying the number of bins for the full-sized NYC taxi and Stack Overflow datasets.
> > A principled and effective way to choose the number of bins is to base the choice on concrete requirements of the application. The exploration of the spike prediction task supports this opinion.
> >
> > > **Generalization boundaries: For which types of datasets (e.g., continuous vs. discrete, unimodal vs. multimodal) do you expect the categorical output to perform best? Have you tested cases where discretization could harm performance?**
> >
> > The categorical output will perform best (comparatively) in the failure modes of other approaches. Output heads that cannot represent many modes (const, exp, softplus) will perform poorly on multi-modal datasets. Mixtures are susceptible to instability due to singularities. The categorical output utilizes the training set to estimate distribution quantiles, so it can suffer when training sets are very small, which is demonstrated in Figure 3, 6, 11-16.
> >
> >
> > > **Computational aspects: How does the categorical formulation affect training speed and memory compared to mixture-based models? Based on your experience, is the categorical head not only more stable but also more efficient?**
> >
> > Taking the `gpt-a-cat` and `gpt-a-logmix` as examples, there is little difference in batches/second. Weights are 1 x n_bins for the categorical head vs 3 x n_components for the lognormal mixture head; for the `gpt-a` base model, the head weights are a small fraction of the total. The training curves are highly dependent on the datasets, including training set size. If not in an overfitting regime, instability of the lognormal mixture may place an upper bound on convergent loss.
> >
> > > **Reproducibility: You mention that supplementary code and datasets are provided for all experiments, but I was unable to find a repository link or reference in the submission. Could you kindly clarify where the code and data will be available?**
> >
> > The code is available in the supplementary material.

---

> > > ### Comment · Reviewer_uxbv · 2025-11-24
> > >
> > > Thank you for the detailed response. I prefer to keep the score as it is, given the problems mentioned and the form of the paper at the time of the submission.

---

### Note · Authors · 2025-12-01

I have read and agree with the venue's withdrawal policy on behalf of myself and my co-authors.